# Was the Last Bacterial Common Ancestor a Monoderm after All?

**DOI:** 10.3390/genes13020376

**Published:** 2022-02-18

**Authors:** Raphaël R. Léonard, Eric Sauvage, Valérian Lupo, Amandine Perrin, Damien Sirjacobs, Paulette Charlier, Frédéric Kerff, Denis Baurain

**Affiliations:** 1InBioS–Centre d’Ingénierie des Protéines, Université de Liège, 4000 Liege, Belgium; rleonard@doct.uliege.be (R.R.L.); ericsauvage8@gmail.com (E.S.); valerian.lupo@doct.uliege.be (V.L.); paulette.charlier@uliege.be (P.C.); 2InBioS–PhytoSYSTEMS, Unit of Eukaryotic Phylogenomics, Université de Liège, 4000 Liege, Belgium; d.sirjacobs@uliege.be; 3University Lille, CNRS, Centrale Lille, UMR 9189 CRIStAL, F-59000 Lille, France; amandine.perrin@pasteur.fr; 4Hub de Bioinformatique et Biostatistique-Département Biologie Computationnelle, Institut Pasteur, 75015 Paris, France

**Keywords:** bacterial evolution, cell-wall, outer membrane (OM), Bayesian inference (BI), phylogenomics, comparative genomics, ancestral traits

## Abstract

The very nature of the last bacterial common ancestor (LBCA), in particular the characteristics of its cell wall, is a critical issue to understand the evolution of life on earth. Although knowledge of the relationships between bacterial phyla has made progress with the advent of phylogenomics, many questions remain, including on the appearance or disappearance of the outer membrane of diderm bacteria (also called Gram-negative bacteria). The phylogenetic transition between monoderm (Gram-positive bacteria) and diderm bacteria, and the associated peptidoglycan expansion or reduction, requires clarification. Herein, using a phylogenomic tree of cultivated and characterized bacteria as an evolutionary framework and a literature review of their cell-wall characteristics, we used Bayesian ancestral state reconstruction to infer the cell-wall architecture of the LBCA. With the same phylogenomic tree, we further revisited the evolution of the division and cell-wall synthesis (*dcw*) gene cluster using homology- and model-based methods. Finally, extensive similarity searches were carried out to determine the phylogenetic distribution of the genes involved with the biosynthesis of the outer membrane in diderm bacteria. Quite unexpectedly, our analyses suggest that all cultivated and characterized bacteria might have evolved from a common ancestor with a monoderm cell-wall architecture. If true, this would indicate that the appearance of the outer membrane was not a unique event and that selective forces have led to the repeated adoption of such an architecture. Due to the lack of phenotypic information, our methodology cannot be applied to all extant bacteria. Consequently, our conclusion might change once enough information is made available to allow the use of an even more diverse organism selection.

## 1. Introduction

Cell-wall architecture has always been an important morphological character for bacterial classification [1]. Two main types of cell wall exist: the monoderm and the diderm architectures. While monoderm bacteria are generally surrounded by a thick peptidoglycan (and are positive to Gram coloration), in diderm bacteria, a thin peptidoglycan layer is sandwiched between the cytoplasmic membrane and the outer membrane (and are negative to Gram coloration) [2,3]. However, cell-wall features are insufficient to yield a classification that would correlate with phylogenetic trees based on molecular data [4]. Hence, distantly related phyla may have apparently identical cell walls (e.g., Negativicutes and Proteobacteria), whereas closely related phyla or families may present variations in their peptidoglycan thickness or composition, and even in the number of surrounding membranes (e.g., Negativicutes and Halanaerobiales compared to other Firmicutes) [5]. Nonetheless, the evolution of the bacterial cell wall should be addressed considering the phylogeny of the domain. The number of membranes (one or two) that surround a bacterial cell, their lipid composition and the thickness of the peptidoglycan layer are undoubtedly major characteristics of the bacterial cell wall, and these features often come into consideration when discussing the evolution of the bacterial domain. Hence, transition from one to two lipid membranes (or the opposite) has attracted much attention. Disappearance of the outer membrane going from “diderm” to “monoderm” architecture has been proposed by Cavalier-Smith [6,7] but evolution from monoderm to diderm bacteria is usually favoured by other evolutionary biologists [8,9,10,11]. It has been suggested that the endosymbiosis between an “actinobacterium” and a “clostridium” could be the starting point for the onset of double-membrane bacteria [12], but how exactly this symbiosis could have further evolved to form a diderm bacterium still is to be detailed. An attractive hypothesis accounting for the emergence of the outer membrane is its evolution from a forespore of a spore-former “firmicute”. Based on 3D electron cryotomographic images of spore formation in the diderm firmicute *Acetonema longum*, Tocheva et al. showed that the inner membrane (IM) of the mother cell is inverted to become the outer membrane of the forespore and ultimately of the germinating cell [13], leading to the assumption that the outer membrane of diderm bacteria could have evolved from monoderms via sporulation [11,13,14,15]. In contrast, some studies of the evolution of the cell-wall architecture in the phylum Firmicutes interpreted the double membrane found in Halanaerobiales and Negativicutes (two classes of Firmicutes) as a reminiscence of the double membrane in the Firmicutes ancestor, and thus concluded that the outer membrane was lost multiple times in this phylum [16,17]. This interpretation further opens the possibility that the last bacterial common ancestor (LBCA) was a bona fide diderm bacterium.

Cell division in bacteria involves a series of proteins that fulfil many functions as diverse as cytoplasmic membrane invagination, DNA transfer control, peptidoglycan synthesis and daughter cell separation. They assemble into a dynamical complex that overpasses the cytoplasmic membrane and has components in both the cytoplasm and the periplasm. A small number of these proteins are essential and conserved in the genome of almost all bacteria [18]. Several of these proteins of cell division are generally clustered together with proteins involved in peptidoglycan synthesis in a single locus on the genome, the *dcw* (division and cell-wall synthesis) cluster [18]. This cluster is found in many bacteria and its composition and gene order are generally well conserved [19,20]. It has also been shown to be one of the most stable gene clusters (the cluster itself and the gene synteny within the cluster are conserved in a broad taxonomic range of genomes) [18], on par with the ribosomal clusters [21,22]. The longest version of the *dcw* cluster includes 17 genes and encompasses genes coding for proteins responsible for peptidoglycan precursors synthesis (DdlB, MurA, MurB, MurC, MurD, MurE, MurF, MurG, MraY), proteins integrated in the divisome (FtsA, FtsI, FtsL, FtsQ, FtsW, FtsZ), and proteins involved in regulation via DNA binding or RNA methylation (MraW, MraZ). The *E. coli dcw* cluster includes 15 genes, starting with *mraZ* and ending with *ftsZ*, but misses the *murA* and *murB* genes [23]. Many phyla, orders, classes, or families are apparently characterized by the lack of specific genes in the cluster, the absence of *ftsA* and *ftsZ* in Chlamydiae and Planctomycetes being well-known examples [24]. These observations suggest that the organization of the *dcw* cluster holds clues to bacterial evolution. Thus, its detailed study might complement sequence-based phylogenomic approaches, including in terms of rooting of the bacterial tree. For example, the integration of a gene in a specific position within the cluster probably happened only once in the history of the bacterial domain, whereas gene loss and genomic reorganization events, on the contrary, are expected to have been more frequent. Likewise, the phylogenetic distribution of the genes involved in the biosynthesis of the outer membrane in diderm bacteria might provide useful information about their evolutionary status, ancestral or derived, with respect to the bacterial domain as a whole [5,17,25].

In this work, we built a Bayesian phylogenomic tree of the bacterial domain using a supermatrix of 117 single-copy orthologous genes sampled from 85 species representative of the bacterial diversity and for which a descriptive literature exists. We then researched the cell-wall architectures for these species and used the tree to reconstruct the evolution of two cell-wall traits, the number of membranes and the presence and thickness of the peptidoglycan layer, again with Bayesian inference. Moreover, we compared the composition and gene order of the *dcw* cluster in our 85 representative species and used a new variant of a homology-based method to map the organization of the *dcw* cluster on the evolution of the bacterial domain. Contrary to our expectations based on recent literature and educated guesses, our Bayesian analyses inferred that the LBCA was a monoderm bacterium with a thick peptidoglycan. This reconstruction implies that the outer membrane of diderm bacteria appeared more than once, a hypothesis that is indeed supported by differences in the genetic machinery involved in its biosynthesis across the various diderm lineages, as shown by our extensive similarity searches. Our results also show that the LBCA already had a complete *dcw* cluster and that its organization does not correlate with cell-wall architecture.

## 2. Materials and Methods

### 2.1. Dataset Assembly

#### 2.1.1. Data Download

The initial dataset of prokaryotic genomes and proteomes was downloaded from Ensembl Bacteria release 20 [26] using wget. This dataset had 8848 Bacteria and 238 Archaea represented.

#### 2.1.2. Genome Dereplication and Selection

We first reduced the number of genomes based on genomic signatures [27] to regroup similar genomes into genome clusters with a prerelease version of our new software ToRQuEMaDA [28]. Briefly, for five different k-mer sizes (from 2 to 6-nt), we computed the frequency of each word in each genome using the program compseq from the EMBOSS software package [29]. The complete lineage of every genome was recovered from the NCBI Taxonomy database [30] using the program fetch-tax.pl from the Bio::MUST::Core distribution (D. Baurain, https://metacpan.org/dist/Bio-MUST-Core, accessed on 16 February 2022). Each signature file was further analysed in R [31] to cluster genomes into a predefined number of groups (300, 600, 900, 1200, 1500 and 2100) using various distance metrics (i.e., Euclidean, Pearson and Hamming) and clustering algorithms (i.e., k-means, ascending and descending hierarchical clusterings). To choose the best combination of methods and parameters, the available taxonomic information was used to evaluate the quality of the clustering. Briefly, we computed how many different taxa of each rank (phylum, class, order, family, genus, species) were found in each individual cluster or each set of clusters and chose the combination that best separated the higher-level taxa (phylum, class, order, family) while merging the lower-level taxa (genus, species) [28]. This led us to settle on the following set of methods and parameters: 6-nt k-mer, 900 clusters, Pearson distance and ascending hierarchical clustering algorithm. Then, we selected a single representative for each cluster, based on the quality of genome annotations, as evaluated by the number of gene names devoid of uninformative words like “hypothetical”, “putative”, “unknown” etc [28]. After including a few other well-characterized genomes (e.g., *Streptomyces coelicolor* A3(2), *Escherichia coli* O127:H6 str. E2348/69, *Staphylococcus aureus* subsp. *aureus* MRSA252), we ended up with a list of 903 genomes: 822 Bacteria and 81 Archaea.

#### 2.1.3. Identification of Orthologous Groups

For every protein sequence of every one of these 903 genomes, we launched an all-versus-all BLAST-like similarity search using USEARCH v7.0.959 [32] with the following parameters (evalue = 1 × 10^−5^; accel = 1; threads = 64). Then, we used OrthoMCL v2.0.3 [33] to cluster protein sequences into orthologous groups based on USEARCH reports, using an e-value cut-off of 1 × 10^−5^, a similarity cut-off of 50% and an inflation parameter of 1.5. The total number of proteins for the 903 genomes was 2,467,263, and these were partitioned into 124,422 orthologous groups, whereas 326,269 sequences were considered as “singletons” by OrthoMCL (i.e., without homologues).

#### 2.1.4. Database Creation

Gene metadata (organism, genomic coordinates, strand, putative function) for every protein was extracted from the definition lines of the Ensembl FASTA files and stored into a custom designed MySQL (Oracle Corporation) relational database (see Appendix A), along with orthology relationships, based on our protein sequence clustering.

### 2.2. Evolution of the Bacterial Domain

#### 2.2.1. Supermatrix Assembly

To build a robust tree of the bacterial domain, we manually chose a subset of 85 genomes (out of the 903 genomes initially selected), trying to maximize the number of classes. Then, using classify-mcl-out.pl [34], we selected all orthologous groups of proteins featuring at least one representative of eight major bacterial phyla (Firmicutes, Chloroflexi, Actinobacteria, Deinococcus-Thermus, Proteobacteria, Spirochaetes, Planctomycetes and Bacteroidetes) and in which at most 10% of the selected genomes had more than one gene copy. This left us with a list of 176 broadly conserved and (mostly) single-copy genes. The final dataset was further reduced to 117 orthologous groups to ensure a maximum of 14 missing species in each individual orthologous group (Appendix A). The corresponding orthologous groups were aligned with MAFFT v7.127b [35] using default parameters. The protein sequence alignments were then filtered with Gblocks v0.91b [36] using a set of “medium stringency” parameters (as predefined in Bio::MUST::Core) and concatenated with SCaFoS v1.30k [37]. Finally, the resulting concatenation was further filtered for sites >50% missing character states, yielding a supermatrix of 85 species × 19,959 unambiguously aligned amino-acid (AA) positions (4.29% missing character states). A preliminary (more diverse) supermatrix was also created in the process, including 101 species and 19,959 unambiguously aligned AA positions (4.72% missing states).

#### 2.2.2. Phylogenomic Analyses

For Bayesian inference (BI), we used PhyloBayes MPI v1.5 [38] to produce six replicate Markov Chain Monte–Carlo (MCMC) chains of 50,000 cycles, with one tree sampled every 10 cycles, using the CAT+GTR+Γ model of sequence evolution [39,40,41]. Constant sites were deleted with the -dc option. Convergence was assessed using the program tracecomp from the PhyloBayes software package. Two consensus trees (along with their posterior probabilities) were extracted after a burn-in of 10,000 cycles: one over the six chains (A to F) and another over the two most congruent chains (A and C; maxdiff = 0.130; meandiff = 0.001), both with the -c option of bpcomp set to 0.01. Cross-validation tests to decide the best-fit model (CAT+GTR+Γ) were carried out using PhyloBayes v3.3f [42], as suggested in PhyloBayes manual (p. 38). For our preliminary tree, we ran two chains of 50,000 cycles, with one tree sampled every 10 cycles, under the simpler CAT+Γ model. The consensus tree was extracted after a burn-in of 5000 cycles (maxdiff = 0.580; meandiff = 0.011). All trees (including those described below) were formatted semi-automatically using the scripts format-tree.pl, export-itol.pl and import-itol.pl (also from Bio::MUST::Core) and iTOL v6 [43].

#### 2.2.3. Congruence Tests

Congruence tests were performed on the 85-species supermatrix genes with Phylo-MCOA v1.4 [44], then Maximum Likelihood (ML) reconstruction with RAxML v8.1.17 [45] was used under the model PROTGAMMALGF (LG+F+Γ) to compare the topologies obtained with and without the “cell-by-cell outliers” (i.e., specific species in specific genes whose position is not concordant with their position in the other gene trees) found by Phylo-MCOA.

### 2.3. Evolution of the Cell-Wall

#### 2.3.1. Cell-Wall Architecture of Extant Organisms

For each one of the 85 bacterial species, a dedicated survey of the literature was conducted (Appendix A). When no information about the cell-wall architecture was available at the species level, we searched at a higher taxonomic level, sometimes up to the phylum. Based on the collected data, we summarized the cell-wall architecture using two different traits: the number of membranes and the presence and thickness of the peptidoglycan layer (Appendix A). For the membrane trait, we used the following binary coding: 0 for one membrane and 1 for two membranes, whereas for the peptidoglycan trait, we used three different states: 0 for no peptidoglycan, 1 for a thin peptidoglycan and 2 for a thick peptidoglycan. Cell-wall trait analyses were then performed using BayesTraits V3 [46,47,48]. For *Parachlamydia acanthamoebae*, no clue about peptidoglycan thickness was found, so this trait was coded as “12”, following the suggestion in BayesTraits manual (p. 9).

#### 2.3.2. Correlation between Cell-Wall Traits

Correlation between cell-wall traits was tested by comparing the discrete independent and discrete dependent models using Bayes Factors (BF), as described in BayesTraits manual (p. 13). We applied the steppingstone sampler, using 100 stones with 10,000 iterations per stone. As this procedure only allows for the comparison of two binary traits, and as our peptidoglycan trait had three possible states, we had to combine two different states into a single state. Three different combinations were tested to check the robustness of the correlation. For case A, the absence of peptidoglycan was coded as 0 and the presence of peptidoglycan (either thin or thick) as 1. For case B, both the absence of peptidoglycan and the thin peptidoglycan were coded as 0, while the thick peptidoglycan was coded as 1. For case C, both the absence of peptidoglycan and the thick peptidoglycan were coded as 0, while the thin peptidoglycan was coded as 1. Because *P. acanthamoebae* is a Chlamydiae, which belong to the diderm-LPS group, its undocumented peptidoglycan layer (see above) was considered as thin when recoding the peptidoglycan trait.

#### 2.3.3. Ancestral State Reconstruction of Cell-Wall Traits

For ancestral state reconstruction, the two traits were considered separately. We used the Bayesian phylogenomic tree rooted on Terrabacteria as an input tree, and further checked the robustness of our inferences to five alternative roots, all within Terrabacteria. Branch lengths were scaled to have a mean of 0.1, as suggested in BayesTraits manual (p. 10). Five different MultiState models were tested: prior exponential of 10 (model “E”), hyperprior exponential 0 to 10 (model “H1”), hyperprior exponential 0 to 100 (model “H2”), reverse-jump hyperprior exponential 0 to 10 (model “R1”), and reverse-jump hyperprior exponential 0 to 100 (model “R2”). Reversible-jump models had the opportunity to forbid some transitions (rate = 0) and/or to equate distinct rates. Ten MCMC chains were run for each combination of trait/root/model for 1,100,000 cycles, with one sample saved every 1000 cycles, and burnin set at 100,000 cycles. State probabilities and transition rates were summarized as means of the 10 × 10,000 samples. To investigate the sensitivity of the Bayesian inference of a monoderm LBCA to priors, one more analysis (biased on purpose towards reversion from diderm to monoderm state) was re-run as 100 MCMC chains with q01 and q10 exponential hyperpriors set to 0 to 1 for and 1 to 10, respectively.

#### 2.3.4. Comparison of the Selected Models

Building on the steppingstones sampler files produced by the BayesTraits ancestral state reconstruction, we compared the fit of our five models (in a systematic pairwise fashion) to both the membrane and the peptidoglycan data (used for the ancestral state reconstruction) using Bayes Factors. We selected the steppingstones files from the runs with the tree rooted on the Terrabacteria. As above, the steppingstone sampler used 100 stones with 10,000 iterations per stone.

### 2.4. Evolution of the dcw Cluster

#### 2.4.1. Synteny Analyses of Extant Genomes

To study the gene order of the *dcw* cluster across our 903 genomes, we developed a custom R script. This interactive interface allowed us to select any subset of genomes and to focus on any region of the bacterial chromosome chosen as the reference genome for the comparison. To maximize the robustness of these analyses, the data (genomic coordinates, orthology relationships, functions) needed for the visualization are fetched in real-time from the relational database. Examples of graphical outputs produced by this program (limited to the 85 final organisms) are shown in “synteny_85_dcw.pdf” available in the folder ProCARs. The orthologous groups corresponding to the genes of the *dcw* cluster were identified by a combination of homology searches using reference protein sequences as queries and our R interface for visual confirmation of synteny conservation. In most cases but the poorly conserved *ftsL* and *ftsQ*, a single orthologous group was found for each gene. For *ftsL* and, to a much lesser extent, *ftsQ*, several orthologous groups had to be merged, based on the presence of an unidentified gene sequence at their respective expected location, i.e., between *mraW* and *ftsI* for *ftsL*, and just before *ftsA* for *ftsQ*. Moreover, HMM profiles (pHMM) [49,50] (see also below) were built from unambiguous reference sequences to ensure proper identification of *ftsL* and *ftsQ* genes in genomes with a fragmented *dcw* cluster. Overall, *ftsL* and *ftsQ* were spread over 36 and 24 orthologous groups (many having only 2–3 sequences), respectively, whereas *mraW*, *mraZ* and *ftsA* were spread over 2, 3 and 4 orthologous groups, respectively.

#### 2.4.2. Ancestral Gene Order Reconstruction

To reconstruct the evolution of the *dcw* cluster, we used the program ProCARs [51], modified to prevent gene inversions in the cluster (by enabling the -p option). ProCARs input files were built semi-automatically from the relational database, focusing on the 85 bacterial species of our phylogenomic analyses and informed by synteny analyses of extant genomes. Briefly, genes too far from other genes were encoded as lying on different “chromosomes” by introducing artificial telomeres. When several “orthologous” genes were available in a given genome for a specific gene, we first tried to select the gene copy lying on the artificial “chromosome” with the highest count of other *dcw* genes. If this failed due to ties, we turned to the gene copy located on the main DNA molecule (genuine chromosome or largest scaffold in the genome assembly); otherwise, as a last resort, we selected the gene copy in the same orientation as the *dcw* genes found on the genuine chromosome or largest scaffold. Finally, when two gene copies were in tandem, we considered them as a single (duplicated) gene for the purpose of the ancestral reconstruction.

#### 2.4.3. Phylogenetic Analyses

For the single-gene analyses of the *dcw* cluster in the 85 genomes of interest, we used the 17 identified orthologous groups (possibly merged; see above) to produce trees according to two different approaches: (1) by ML using RAxML v8.1.17 under the PROTGAMMALGF (LG+F+Γ) model and (2) by BI using PhyloBayes v3.3f under the model GTR+C60+Γ, with two MCMC chains run for 10,000 cycles, with burnin of 5000 cycles and sampling every 10 cycles. Convergence was assessed as above (gene maxdiff’s ranging between 0.208 and 1.000 and meandiff’s between 0.013 and 0.062), with the -c option of bpcomp set to 0.25, which turned unresolved nodes to multifurcations. Then, a concatenation of 15 of the 17 genes of the *dcw* cluster was built using SCaFoS v1.30k, leaving out *ftsL* and *ftsQ* due to their poor conservation (see above). For these 15 genes, additional steps were carried out to ensure the orthology of the concatenated sequences. Briefly, we used our ProCARs input to select only the genes belonging to the *dcw* cluster (or sub-cluster) in each genome. Orthologues not supported by synteny evidence were removed from the alignments using prune-ali.pl (also from Bio::MUST::Core) before concatenation. We further filtered out sites with ≥50% missing character states, thereby yielding a sparser supermatrix of 85 species × 4571 AAs (8.47% missing character states). PhyloBayes MPI v1.4 was used to run two chains under the CAT+Γ model for 50,000 cycles. We chose a burnin of 10,000 cycles and kept only one sample every 10 cycles of the remaining 40,000 cycles. We selected both chains to compute the tree (maxdiff = 0.284; meandiff = 0.007), with the -c option of bpcomp set to 0.25. All trees were formatted as above. 

### 2.5. Evolution of the Genes Related to the Outer Membrane

#### 2.5.1. Homology Searches in Complete Proteomes

For our broader study of the taxonomic distribution of 16 genes involved in synthesis and in maintaining the integrity of the outer membrane across the 903 selected genomes (including previously discarded organisms like Thermotogae), we did not rely on synteny as those were not part of a single cluster in any organism. Instead, we searched for the orthologous groups containing unambiguous reference sequences for these genes. For each set of orthologous groups potentially corresponding to a gene of interest (merging from one to nine orthologous groups per gene), we computed an alignment over all sequences with MAFFT v7.453 (using the accurate LINSI strategy) and checked by eye if it was globally satisfactory or not, possibly after cleaning up a few divergent sequences. If the alignment was good enough, we built an HMM profile from it to search the complete proteomes of our 903 genomes using HMMER [49,50]. Then, based on the E-value, length, pHMM profile coverage, copy number and taxonomy of the HMMER hits, we selected the probably orthologous proteins using the visual software Ompa-Pa (A.R. Bertrand and D. Baurain; available at https://metacpan.org/dist/Bio-MUST-Apps-OmpaPa, accessed on 16 February 2022). In contrast, when the alignment of all sequences was too poor, we focused on the original orthologous group containing the *E. coli* sequence and tried to build a profile by adding up to 6 (for *lolB* and *lptC*) of the additional orthologous groups using an iterative strategy as implemented in the software Two-Scalp (A.R. Bertrand and D. Baurain; available at https://metacpan.org/dist/Bio-MUST-Apps-TwoScalp, 16 February 2022). Then, we followed the same route as if the pHMM had been computed from a “good-enough” alignment. For the specific case of the *bamA* gene, we first collected 28 orthologous groups containing proteins annotated as BamA, Omp85 and/or TspB, then we used InterProScan v.5.48-83.0 with default parameters and disabled use of the precalculated match lookup [52] to determine the number of POTRA domains [53] in the 1425 individual sequences. Two curated alignments based on preliminary ML trees (see below) were built: one from the five orthologous groups where the sequences mostly had 4 or 5 POTRA domains (Appendix A), which we considered as the orthologues of the genuine BamA protein of true diderms-LPS, and one from five orthologous groups having 2 or 3 POTRA domains, which included the BamA “4” sequences of Cyanobacteria, as well as related proteins (i.e., BamA-like/Lipo/TamA) [54]. By “curation”, we mean elimination of incomplete and/or divergent individual sequences but without discarding representatives of scarcer groups. Finally, these two alignments were used to build two pHMM profiles and perform HMMER searches as described above.

#### 2.5.2. Taxonomic and Phylogenetic Analyses

For each gene of the 16 genes, we retrieved the list of genomes containing the (probably) orthologous proteins and tabulated the corresponding organisms at the phylum level. From these numbers, we tried to identify recurring patterns of gene distribution. For two genes, *tolA* and *ybgF*, the taxonomic distribution was discordant with respect to other genes (when present) in the atypical diderms group. In each case, only one of the expected phyla of the atypical diderms group had at least a copy, and this phylum was represented by a noticeably lower number of sequences compared to other genes present in the atypical diderms group (when they had copies of the gene). To decide if these discordances were due to genome contamination or very recent gene transfers, we aligned the sequences with MAFFT v7.453 (LINSI) and computed two phylogenetic trees using RAxML v8.1.17 under the PROTGAMMALGF (LG+F+Γ) model. Trees were also produced for the 14 other genes associated with the outer membrane following the same method. All trees were formatted as above, with unresolved nodes (BP < 25%) turned to multifurcations.

## 3. Results

### 3.1. A Robust Tree of the Bacterial Domain

To serve as the base for evolutionary analysis of the cell-wall architecture and reconstruction of the ancestral gene order in the *dcw* cluster, we needed a tree of Bacteria. With the growing availability of fully sequenced genomes, phylogenomics has developed as a discipline using the tools of phylogenetics but applied to tens to hundreds, or even thousands, sequences of broadly conserved genes [55]. Phylogenomic trees can either be inferred from supermatrices of concatenated genes [56] or through combination of single-gene trees into supertrees [57]. Hence, the phylogenomic tree shown in Figure 1 was computed by Bayesian inference based on a dense (4.29% missing character states) supermatrix of 117 single-copy orthologous genes (see Materials and Methods) sampled from 85 representative bacterial genomes with PhyloBayes MPI under the site-heterogeneous CAT+GTR+Γ model (CATegories + Generalised Time-Reversible + Gamma) of sequence evolution [38,39,40,41]. Congruence analyses were run on the 117 individual genes using Phylo-MCOA [44] and did not reveal incongruent genes or species, beyond 62 individual sequences, which might have experienced gene transfer and/or fast evolution. Once discarded, the overall results did not change, as demonstrated by comparing two control trees (i.e., before and after outlier removal) inferred with RAxML under the LG+F+Γ model (see Appendix A). Regarding model selection, cross-validation analyses on four different models confirmed that CAT+GTR+Γ had the best fit to our dataset, followed by CAT+Γ, then GTR+Γ and finally LG+Γ (Appendix A).

Our unrooted tree is in good agreement with most recent concatenating phylogenomic studies aimed at resolving bacterial evolution [58,59,60,61,62,63,64,65,66,67,68]. In particular, we robustly recovered a bipartition of the bacterial lineages composing the Terrabacteria and the “Hydrobacteria” (=Gracilicutes sensu [69]). Within these “megaphyla” first defined by Hedges and Battistuzzi [58], resolution was weaker, as reflected in the lower posterior probabilities at medium phylogenetic depth, whereas phyla and known superphyla (e.g., FBC, for Fibrobacteres-Bacteroidetes-Chlorobi, and PVC, for Planctomycetes-Verrucomicrobia-Chlamydia) were always clearly resolved. In the Terrabacteria, relationships between member lineages slightly varied from run to run (we ran a total of six independent chains, Appendix A), while in the Hydrobacteria (e.g., FBC, PVC, Proteobacteria), Epsilonproteobacteria were occasionally separated from other groups of Proteobacteria (Appendix A). Some additional phyla initially present in our dataset (i.e., Synergistetes, Fusobacteria and Aquificae) were excluded from the tree shown in Figure 1 because they were difficult to robustly position (e.g., due to the chimerical nature of the Aquificae) without bringing more cell-wall architecture diversity (see also [70,71,72]). Likewise, we further discarded the Thermotogae, which are also chimeras [70], even though their toga might be akin to a modified outer membrane [73,74] (see Appendix A for a preliminary 101-species tree including all these lineages). Such uncertainties are not uncommon in bacterial phylogenomics and are the result of a combination of weak phylogenetic signal, widespread lateral gene transfer and systematic error (e.g., long-branch attraction artifacts) [72,75,76,77,78,79,80,81,82].

Rooting the different domains of Life is not an easy issue [82]. In Figure 1, we chose to set the root of Bacteria between Terrabacteria and Hydrobacteria/Gracilicutes, following studies having included Archaea as an outgroup [25,41]. Remarkably, this basal split mirrors cell-wall architecture differences. In the first group, Firmicutes, Tenericutes, Actinobacteria, and presumably Chloroflexi (see below), are mostly monoderm bacteria. Together with the atypical diderms, i.e., Deinococcus-Thermus, Cyanobacteria, Synergistetes and Thermotogae, they compose Terrabacteria [58]. On the other hand, the remaining lineages are diderms mostly featuring lipopolysaccharides (LPS) and correspond to Hydrobacteria/Gracilicutes; these will be called “true diderms-LPS” in this study. Over time, several positions for the bacterial root have been proposed (Appendix A). In the following, because our Bayesian analyses required a rooted tree, we tested several of them, yet excluding roots lying within the true diderms-LPS, which are likely monophyletic (see below). Beyond the root of Figure 1, we thus explored the effect of setting the bacterial root within Terrabacteria on our inferences.

### 3.2. Evolution of the Cell-Wall Architecture

To study the evolution of the cell-wall architecture, we carried out a thorough literature survey on all the bacteria kept in our tree (Appendix A). For each organism, we collected the number of membranes, the presence and thickness of the peptidoglycan layer and, if relevant, the type of spore, as there exists evidence of potential functional connection between sporulation and cell-wall remodelling processes [13,14]. However, preliminary analyses showed that the spore trait was difficult to encode reliably in terms of homologous states. Therefore, it was eventually discarded, whereas the two traits linked to the cell wall itself were analysed using BayesTraits under the MultiState model.

Based on this survey (Appendix A), most bacterial phyla have two membranes (diderm architecture) and a thin peptidoglycan layer. For example, Proteobacteria, Nitrospirae, Acidobacteria, Bacteroidetes and Chlorobi fall into this category and correspond to true diderms-LPS lineages. For the organisms belonging to the PVC superphylum, this architecture might be slightly different [83]. Actinobacteria are essentially monoderms with a thick peptidoglycan, whereas Firmicutes and Chloroflexi both have monoderm and diderm representatives. Firmicutes include Bacilli and Clostridia, two groups of endospore formers. Clostridia and Bacilli correspond to two well-defined classes, sharing many traits though being also very distinct. All Bacilli and most Clostridia are monoderms with a thick peptidoglycan, but some Clostridia [84] (Halanaerobiales and Thermoanaerobacteriales) and the Negativicutes have two membranes (some with lipopolysaccharides in the outer membrane) and a relatively thin peptidoglycan layer [16,85,86]. Regarding the status of the Chloroflexi cell-wall architecture, it is still controversial [68,87,88]. Beside these canonical diderm and monoderm phyla, respectively corresponding to classical Gram- and Gram+ bacteria, there exist a series of organisms with atypical cell-wall architectures. Hence, Deinococcus-Thermus and Cyanobacteria are diderm bacteria with an outer membrane, but their cell walls differ from those of the true diderms-LPS by having a thick peptidoglycan instead of a thin layer (Appendix A).

Consequently, the number of membranes observed in the extant organisms is either one (state 0) or two (i.e., there is an outer membrane, state 1; Appendix A). The evolutionary analysis of this trait suggests a LBCA surrounded by only one membrane. This inference is robust to five model variants (E, H1, H2, R1 and R2; see Materials and Methods) and six different positions for the bacterial root (P(0) = 94.2% to 98.2%; Appendix A). Due to the robustness of our results to alternative rootings, we will only present those obtained with a root located between Terrabacteria and true diderms-LPS (as in Figure 1). In accordance with the inference of a monoderm LBCA, the posterior transition rates indicate that it is easier to gain (q01) an outer membrane (range of the five model’s mean = 2.288–2.495, Table 1) than losing (q10) an existing one (range = 0.008–0.132). If we try to alter the H1/H2 model hyperpriors to promote the loss (q10 = 1–10) at the expense of the gain (q01 = 0–1), the LBCA remains inferred as a monoderm in 67.1% of the cases (mean P(0)), whereas it is inferred as a diderm in 32.9% of the cases (mean P(1)) (Table 1). Concerning the rates, the inferred loss rate remains weak (mean q10 = 0.000–0.187; Table 1), while the distribution of the gain rate (q01) becomes bimodal, with a mode at 0.2 and another at 1.8 (Appendix A) and remain low for the loss rate (q10) (Appendix A). Consequently, under this extreme parameterization, we distinguish two main configurations for the pair of rates (Appendix A) and the monoderm probability P(0) (Appendix A).

In the 85 extant organisms considered in our study, the peptidoglycan layer is either absent (state 0), present and thin (state 1) or present and thick (state 2; Appendix A). The LBCA is inferred with a thick peptidoglycan. While this result is robust to alternative positions of the root, some models (E and H2) let the possibility open (22.0–38.6%, Table 1) for the LBCA having been devoid of peptidoglycan (Appendix A). Moreover, the posterior rates are highly heterogeneous, depending on the transition considered, and present a sensitivity to the model used (mean range = 0.000–20.967; Appendix A and Table 1). Based on the values of the rates, the thin peptidoglycan state (state 1), once acquired, is unlikely to change towards another state, whereas the other two states (states 0 and 2) can exchange freely or change towards the thin peptidoglycan state (Appendix A and Table 1).

In a second step, we used BayesTraits to reconstruct the state of the characters for the Last Common Ancestor (LCA) of every one of the 15 bacterial phyla included in our study, as well as the LCA of several larger groups (e.g., PVC, Terrabacteria), still based on the Terrabacteria root (Figure 2). As expected, the LCA of the true diderms-LPS bacteria is inferred as a diderm organism featuring a thin peptidoglycan layer, whereas the Terrabacteria LCA is reconstructed as a monoderm with thick peptidoglycan. The results obtained for the larger groups are homogeneous across the different models (Appendix A). For Firmicutes, which is the only phylum with some architectural diversity in our dataset, two of the five models (E and H2) do not completely settle on an LCA monoderm with a thick peptidoglycan, and instead do not dismiss an LCA without peptidoglycan (17.6% and 30.1%, respectively; Table 1). Finally, a comparison of the fit of the five models using Bayes Factors (Table 2) showed that model R1 was the best, followed by models R2, H1, E, and finally H2. Therefore, the two models that do not fully agree with the others about the peptidoglycan trait are also those that are deemed less fit by Bayes Factors (E and H2).

Hitherto, the two cell-wall traits were analysed separately, owing to the limitations of the MultiState model used. However, from a biological point of view, their evolution might be correlated. To account for this possibility, we conducted the BayesTraits procedure to estimate the correlation between two traits, which revealed that the peptidoglycan and the membrane characters are indeed linked. The actual strength of the correlation depended on the scheme used to recode the three-state peptidoglycan trait into a binary character, which was needed to estimate the correlation with the membrane trait (see Materials and Methods). When the coding scheme rewarded the mere presence of the peptidoglycan layer, whatever its thickness, the correlation was supported by strong evidence (log Bayes Factor for case A = 9.0), while it raised to very strong evidence when the scheme emphasized either a thick peptidoglycan (case B = 27.6) or a thin peptidoglycan (case C = 37.8). These differences in correlation can easily be explained. In case A, almost all organisms of our study without peptidoglycan are also deprived of the outer membrane (see *Parachlamydia acanthamoebae* in Figure 1), whereas organisms with a peptidoglycan layer often have an outer membrane. In case B, all organisms without peptidoglycan or with a thin peptidoglycan layer are put in the same category. In our study, all organisms with a thin peptidoglycan layer have an outer membrane, and they are more numerous than the organisms without a peptidoglycan layer. In case C, the organisms with a thin peptidoglycan layer have their own category and, in our study, all these organisms also feature an outer membrane.

### 3.3. Evolution of the Gene Order within the dcw Cluster

Initially, we studied the organization of the *dcw* cluster in extant organisms based on the output of a custom visualization software showing orthologous gene groups in their syntenic context (see Materials and Methods for details and “synteny_85_dcw.pdf” available in the folder ProCARs from our Figshare, for the status of the *dcw* cluster in the 85 bacteria of our phylogenomic tree). This approach led us to identify the orthologous groups for the 17 genes of (the most complete form of) the *dcw* cluster. In Cyanobacteria, the nearly total absence of the *dcw* cluster is noteworthy: *mraZ* and *ftsA* are missing from all cyanobacterial genomes examined, and all other genes of the cluster are generally present but completely dispersed on almost as many loci as the number of genes, with some exceptions, the doublet *murC* and *murB* or the doublet *ftsQ* and *ftsZ* (see .xlsx file available in the folder ProCARs). The *murA* gene can be found in clusters or sub-clusters in several genomes. The complete form of the *dcw* cluster is only seen in a single order of Clostridia, the Halanaerobiales (more precisely, in *Acetohalobium arabaticum*). Halanaerobiales are robustly affiliated to Firmicutes yet branching at the root of the phylum [90]. However, *murA* is also present in sub-clusters in Cyanobacteria, Planctomycetes, Lentisphaerae and *Caldithrix abyssi*. Otherwise, if present in the genome, *murA* is usually outside of the *dcw* cluster. Beside this specific gene and particular phyla, several true diderms-LPS phyla are characterized by the loss of specific genes from the cluster (*ftsW* in Thermodesulfobacteria, *murB* and *ddlB* in the FBC superphylum, *ftsA* and *ftsZ* in Chlamydiae and Planctomycetes) (see .xlsx file available in the folder ProCARs).

Taking the rooted phylogenomic tree of Figure 1 as an evolutionary framework and the orthologous groups identified just above as input extant data, we used a new variant of a homology-based reconstruction method (ProCARs) [51] to retrace the evolution of the organization of the *dcw* cluster in our 85 representative organisms. Our reconstruction shows that both the LBCA and the LCA of the Terrabacteria group were organisms featuring a complete 17-gene *dcw* cluster. In contrast, the reconstructed cluster for the ancestor of the true diderms-LPS group included 16 genes, with the *murA* gene outside of the cluster (even if present in the genome). Detailed study revealed that the *murA* gene was also outside of the main cluster in every reconstructed ancestor among true diderms-LPS (Figure 3A). This gene is at best found on a small sub-cluster, and most of the time it exists as a singleton. An example of such a small sub-cluster reconstructed by ProCARs can be seen in the LCA of the FBC superphylum where *murA* and *murB* are in tandem. A parsimonious way to explain these observations would be that the *murA* gene has left the *dcw* gene cluster (but persisted in the genome) of the LCA of true diderms-LPS and the LCA of Actinobacteria, Deinococcus-Thermus and Chloroflexi (assuming these three phyla share a common ancestor). Alternatively, it was lost independently in the three latter phyla. Overall, the *dcw* cluster is conserved in almost all high-level ancestors down to the phyla (see Figure 3A for a summary and .xlsx file available in the folder ProCARs, for details). This conservation mostly takes the form of a single cluster (e.g., Proteobacteria LCA) or of a limited number of sub-clusters, with the synteny retained within individual sub-clusters (e.g., Chloroflexi LCA, Planctomycetes LCA). Thus, the *dcw* cluster appears as an ancient locus with mainly a history of gene loss or gene delocalization, but likely no gene gain since its establishment before the advent of the LBCA.

Phylogenetic trees for the 17 genes of the *dcw* cluster were computed from protein sequences, but these trees are not well resolved (“DCW_17_SG.pdf” available in the folder Trees). Known phyla can be supported by low to high bootstrap proportions (BP: 9–100%) and posterior probabilities (0.3–1.0), while the support is always too low to resolve the relationships between phyla, even though general trends, such as the bipartition between Terrabacteria and true diderms-LPS (Firmicutes–Chloroflexi–Actinobacteria–Deinococcus-Thermus vs. Proteobacteria–FBC–PVC), are observable in several single-gene trees. Moreover, trees inferred from genes often found outside of the *dcw* cluster (e.g., *murC*, *murB* and *ddlB*) are blurrier than those computed from genes kept in the cluster. Finally, the trees of the genes *ftsQ* and *ftsL*, for which the orthologous groups had to be manually reconstructed (see Materials and Methods) are particularly chaotic. In contrast, the *mraY* tree (Appendix A) is better supported (BP: 39–100%; posterior probabilities: 0.5–1.0) at the phylum level and is the most congruent with the tree resulting from the 117-gene supermatrix (Figure 1). When concatenated, the *dcw* genes (all but *ftsQ* and *ftsL*) recover a similar tree (Appendix A), notably featuring the Terrabacteria group, the FBC group and the true diderms-LPS, but with one exception: the PVC group is split in three, with the Planctomycetes and Verrucomicrobia on one side, the Chlamydia on the other side and the Lentisphaerae within the FBC group. This suggests that the *dcw* cluster mostly experienced a vertical evolution.

### 3.4. Evolution of the Genes Related to the Outer Membrane

According to our ancestral reconstruction of the cell wall, the LBCA had a single membrane around its cell, which implies that the atypical diderms lineages within Terrabacteria (Cyanobacteria, Deinococcus-Thermus and some Firmicutes, i.e., the Halanaerobiales and the Negativicutes) had to acquire their outer membrane independently and in distinct events from the event at the origin of true diderms-LPS. At face value, this inference might seem less parsimonious than hypothesizing a diderm LBCA and multiple independent outer membrane losses over the evolution of the bacterial domain, as suggested repeatedly [5,25,68]. To decide whether the outer membrane could indeed have evolved several times independently, we studied the taxonomic distribution of 16 genes involved in outer membrane synthesis and integrity: *bamA*, *lolB*, *lptA*, *lptB*, *lptC*, *lptD*, *lptE*, *lptF*, *lptG*, *pal*, *tolA*, *tolB*, *tolQ*, *tolR*, *ybgC*, *ybgF*. Briefly, BamA is the main protein of the Bam complex (to which the other Bam proteins attach to), which is responsible for the assembly of beta-barrel proteins in the outer membrane [91]. LolB is the only outer membrane-anchored protein of the Lol pathway, which delivers lipoproteins to the outer membrane [3]. The Lpt system (LptA to LptG) ensures the transport of the lipopolysaccharides from the cytoplasm to the outer membrane [92]. Finally, the Tol-Pal system (Pal, TolA, TolB, TolQ, TolR, YbgC, YbgF) is involved in the uptake of colicin, the uptake of filamentous bacteriophage DNA and the integrity of the outer membrane [93].

The distribution of these genes was examined across our first selection of 903 bacterial genomes (all genomes even the previously discarded ones) using curated Hidden Markov Model (HMM) profiles built from orthologous groups including *E. coli* reference sequences and complemented by phylogenetic analyses when orthology was doubtful (see Materials and Methods for details). These results were then summarized at the phylum level to identify recurring patterns of gene distribution (Figure 3B and “OM_genes_presence-hmms.csv” available in the folder Outer_membrane, for details), while single-gene trees inferred from the corresponding protein sequences are available (“LBCA_OM_16_SG.pdf” available in the folder Trees). Altogether, our study of the genes encoding the proteins BamA, LolB, the Lpt system and the Tol-Pal system revealed four different patterns of presence/absence in bacterial phyla with diderm organisms. These four gene distribution patterns correspond to: (1) “atypical diderms” (see references in Appendix A), i.e., Cyanobacteria, Deinococcus-Thermus and diderm Firmicutes; (2) “monoderm Terrabacteria”, i.e., Chloroflexi, of which some may be monoderms but all are devoid of lipopolysaccharides [68,87], Actinobacteria, and monoderm Firmicutes; (3) “true diderms with LPS” (TDL = typical Gram–bacteria); (4) Thermotogae, in which the outer membrane has been replaced by a toga made of structural proteins and polysaccharide hydrolases (xylanases) [73,74,94]. Below, we briefly comment on these gene distributions from a functional perspective.

First, according to our comprehensive homology searches, *bamA* is exclusive to true diderms-LPS, Deinococcus-Thermus and Thermotogae, even though the latter lack nearly all other outer membrane-related genes studied here. This result suggests a true diderms-LPS origin for Thermotogae, which are now considered as chimeras partly derived from (or at least related to) Aquificales [70,72,95]. This chimerical nature of Thermotoga is the reason we did not include them in our phylogenomic tree (see above). Regarding the presence of the *bamA* gene in the atypical diderms of the group Deinococcus-Thermus, it has already been reported [96] and this result appears less compatible with a monoderm LBCA. However, in other atypical diderms, we could not find a genuine BamA protein. Instead, Cyanobacteria and diderm Firmicutes feature proteins that have a quite different domain architecture (see BamA4 and BamA-like in Heinz et al., 2014 [54]) and for which the orthology (i.e., overall sequence similarity due to vertical descent only) with the typical BamA is at best dubious. Therefore, we currently disagree with the idea that BamA per se would be common outside true diderms-LPS [97]. Nonetheless, BamA, taken as a family regrouping the typical BamA, “BamA4” and “BamA-like” proteins, might indeed be an essential family (each sub-group sharing a similar function) to all diderm (i.e., featuring an outer membrane) but its members do not necessarily share a vertical transmission from a single ancestral protein. To verify this hypothesis would require a whole new study and is thus not expanded in the current article. Second, *lolB* is exclusive to Proteobacteria, a member of true diderms-LPS, whereas *lptB* (Lpt system) and *ybgC* (Tol-Pal system) are found in all (or almost every) bacterial phylum of our selection of 903 genomes (including Chloroflexi) and are thus not informative about the origins of the outer membrane. It is likely that these two genes have function(s) outside their respective system, functions that could be unrelated to the outer membrane. This has already been proposed for *ybgF*, which might be part of a protein network involved in phospholipid biosynthesis [98]. On the opposite, the LptB protein is known to assemble with LptF and LptG to form an ABC transporter for lipopolysaccharides [92,99], but the two corresponding genes are apparently lacking in Acidobacteria (true diderms-LPS), Tenericutes and Chloroflexi. Perhaps unexpectedly, this is also the case for Actinobacteria, these monoderm bacteria further sharing with Chloroflexi the same distribution pattern for the 16 genes involved with the outer membrane.

Beyond *lptB* and *ybgC*, the Lpt and Tol-Pal systems are found in both atypical diderms and true diderms-LPS but to a different extent. Indeed, both systems are present in atypical diderms, albeit only in a largely reduced form, whereas in true diderms-LPS, they range from a largely reduced form (e.g., Chlamydiae or Planctomycetes) to a (almost) complete form (e.g., Proteobacteria or Bacteroidetes), and this distribution is phylum-specific (Figure 3B). Hence, two genes from each system are only present in (most) true diderms-LPS genomes, *lptD* and *lptE* on one side, *pal* and *tolB* on the other side, whereas all four genes are never found in atypical diderms genomes. Regarding *tolA* and *ybgF*, they may or may not be exclusive to true diderms-LPS, depending on the biological reality of their scarce occurrence in some organisms belonging to atypical diderms (Firmicutes for *tolA* and Cyanobacteria for *ybgF*). Based on our trees of the corresponding proteins, the dubious sequences (denoted by “?” in Figure 3B and by stars in “OM_genes_presence-hmms.csv” available in the folder Outer_membrane) are sisters to Bacteroidetes (member of true diderms-LPS) in both cases, plus one case with a sequence sister to Moraxella in *tolA* tree (Appendix A, see also “LBCA_OM_16_SG.pdf” available in the folder Trees). Therefore, provided they are not the product of genome contamination [100], these genes are unlikely to have been vertically inherited.

From a functional point of view, the genes retained by atypical diderms for the Lpt system (*lptA*, *lptB*, *lptC*, *lptF* and *lptG*) are involved in the transport of the lipopolysaccharides from the cytoplasm to the outer membrane and thus are not directly associated to the outer membrane itself, contrarily to *lptD* and *lptE*, which form a complex at the outer membrane that may serve as the recognition site for the lipopolysaccharides [101]. Similarly, for the Tol-Pal system, atypical diderms genomes lack *pal* and *tolB*, two genes encoding proteins located in the periplasm and therefore directly associated to the outer membrane [102,103]. Overall, the Lpt and Tol-Pal systems in atypical diderms are thus restricted to components that might have a function in the absence of an outer membrane.

Remarkably, the genes of the Tol-Pal system are clustered in most genomes of Proteobacteria and Chlorobi, as well as in the lone genomes we studied within Fibrobacter and Gemmatimonadetes, and sporadically in those of Verrucomicrobia and Acidobacteria (available in the folder Outer_membrane sub-folder synteny_output). As all these lineages belong to the true diderms-LPS, we cannot exclude that the conservation of the Tol-Pal cluster appears patchier than it really is, owing to uneven levels of genome assembly. Regarding the genes of the Lpt system, they are not clustered in any of the genomes examined, except in Proteobacteria, where five of the seven genes are grouped on two loci (*lptFG* and *lptABC*) (available in the folder Outer_membrane sub-folder synteny_output). Nevertheless, as the synteny of the genes of both Lpt and Tol-Pal systems was only studied in the 85 genomes of our phylogenomic tree, we may have missed non-Proteobacterial genomes in which some of the *lpt* genes are indeed clustered, as reported in the recent study of Taib et al. [17].

## 4. Discussion

The nature of the LBCA is unknown, especially the architecture of its cell wall. The lack of reliably affiliated bacterial fossils outside Cyanobacteria [104] makes it elusive to decide the very nature of the LBCA. Nevertheless, phylogenomic inference leads to informative results, and our analysis of the cell-wall characteristics of extant bacteria, combined with ancestral state reconstruction and distribution of key genes, opens interesting possibilities: the LBCA might have been a monoderm bacterium featuring a complete 17-gene *dcw* cluster, two genes more than in the model *E. coli* cluster. This result was also supported by the recent study of [105], in which the authors found 146 protein families that formed a predicted core for the metabolic network of the LBCA. From these families, phylogenetic trees were produced and the divergence of the modern genomes from the root to the tips was analysed. It appears that the Clostridia (a class of Firmicutes) are the least diverged of the modern genomes and thus the first lineage to diverge from the predicted LBCA were similar to the modern Clostridia. Based on these results, the authors suggested that the LBCA could have been a monoderm bacteria.

As diderm bacteria are not monophyletic, whatever the root used for the bacterial domain, our reconstruction of a monoderm LBCA implies that the diderm character state has appeared several times, which goes against the principle of parsimony commonly invoked in such matters [68]. Indeed, acquiring an outer membrane is more than a simple mutation: it requires the acquisition of a whole new complex system. This makes the “monoderm-first” result counter-intuitive to the opposite of the alternative, widely held “diderm-first” hypothesis, in which the outer membrane is an ancestral feature having evolved only once in the LBCA and later lost in monoderms [5,17,25,68]. However, such an observation can be made in Archaea, where most of the studied organisms have a monoderm cell wall featuring a S-layer and/or pseudomurein, methanochondroitin and protein sheaths. In this context, some diderm Archaea have been reported in different distant phyla, like the Crenarchaeon *Ignicoccus hospitalis*, the Euryarchaeon ARMAN (Archaeal Richmond Mine Acidophilic Nanoorganisms) or the *Candidatus Altiarchaeum hamiconnexum* (SM1 Euryarchaeum) in the DPANN group [106]. Although it has not been proved that a monoderm cell wall is the general architecture in Archaea, the discovery of diderm Archaea within different phyla shows that acquisition of a second membrane has occurred multiple times during archaeal evolution. Moreover, our results are model-based, congruent across different roots and models and robust to a heavily biased hyperprior towards the diderm-first hypothesis. It contrasts with other recent studies, which do not rely on probabilistic models [5,68] and conclude to a diderm LBCA, based on qualitative considerations. That being said, the diderm-first view has also been supported in the recent work of Coleman et al. [25]. The latter study featured a reconciliation tree and infered the diderm state of the LBCA based on the genes involved in lipopolysaccharides synthesis and the flagellar subunits, notably PilQ, which is part of the Type IV pili. While the approach of Coleman and co-workers was also model-based, it differed from ours by first inferring the gene catalogue of the LBCA and then deducing its cell-wall architecture, whereas we directly infer the LBCA architecture and then studied the underlying gene distribution patterns to corroborate our inference. It is of note that the Type IV pili is also present in monoderm bacteria [107], thus its presence does not automatically entail the inference of a diderm LBCA.

Hence, following a bibliographic search for proteins with functions exclusive to diderms (without distinguishing between diderms with and without lipopolysaccharides), we identified 16 candidates: BamA, which is part of a complex assembling the proteins in the outer membrane [91], LolB, which is part of the proteins inserting the lipopolysaccharides in the outer membrane [3], the Lpt proteins, which serve as a transport chain from the inner, i.e., cytoplasmic [108], membrane (IM) to the outer membrane [92], and the Tol-Pal system, the exact function of which is still unknown but important to the integrity of the outer membrane [93]. Then, we studied the distribution of the 16 corresponding genes in 903 broadly sampled bacterial genomes. Four recurring patterns of outer membrane gene distribution were identified (Figure 3B): (1) atypical diderms (Deinococcus-Thermus, Cyanobacteria and diderm Firmicutes), (2) monoderm Terrabacteria (Actinobacteria, Chloroflexi and monoderm Firmicutes), (3) true diderms-LPS, and (4) Thermotogae. Thermotogae have chimerical genomes [70] and are likely derived with respect to other bacteria; thus, their cell-wall architecture is of secondary origin. Therefore, we do not elaborate further on their case. For similar reasons, the atypical cell-wall of the Corynebacteriales (an order of the Actinobacteria phylum) is not considered in this work. Indeed, Corynebacteriales are positioned deeply within Actinobacteria [109], which again implies a secondary origin for their peculiar cell-wall architecture.

From these patterns, it appears that even monoderm Terrabacteria share some genes involved with the outer membrane despite their lack of an outer membrane. It implies that these genes provide at best circumstantial evidence concerning the presence or the absence of an outer membrane. Thus, solely relying on their detection to infer the presence of an outer membrane would be hazardous. In the study of Coleman et al. [25], the authors build upon two types of genes to justify their inference of a diderm LBCA: the genes involved with the lipopolysaccharides synthesis and the genes involved with the pili type IV. However, our results show that the mere presence of lipopolysaccharides genes is an unreliable feature to infer the presence of an outer membrane, given that even monoderm bacteria can carry some of them. Similarly, the study of [107] showed that the type IV pili is not exclusive to the diderm bacteria. Therefore, the inference of a diderm LBCA by Coleman et al. was based on genes that only provide ambiguous evidence for the outer membrane.

Pattern 2 shows that Chloroflexi share the same gene distribution as monoderm Terrabacteria, despite being mostly considered as diderms (3 out of 4 genomes) in our reconstruction of the cell wall. Currently, there is still debate on whether Chloroflexi are monoderm or diderm organisms, microscopical observations having been inconclusive so far but hinting at the presence of an outer membrane in some Chloroflexi [87,88]. The fact that they share the same outer membrane gene distribution pattern as monoderm Terrabacteria is a clue in favour of Chloroflexi having only one membrane too. In this case, our reconstruction of the LBCA’s cell wall would have had a small bias towards the diderm state and, despite that unwarranted handicap, we still recovered the LBCA as a monoderm bacterium. In our opinion, this result can be taken as more evidence for a genuinely strong signal for a monoderm LBCA.

Patterns 1, 2 and 3 may be arranged following a gradual complexification, with pattern 2 being the simplest, pattern 1 the intermediate and pattern 3 the most complex. The study of the functions of the proteins characterizing the different patterns reveals that pattern 3 is the only one including proteins directly involved with the outer membrane (i.e., linked to the outer membrane), whereas pattern 1 only includes proteins indirectly involved with the outer membrane (i.e., linked to the IM or interacting with the IM or located in the cytoplasm) and pattern 2 only includes proteins indirectly involved with the outer membrane and located in the cytoplasm. Although we know (some of) the outer membrane pathways functioning in true diderms-LPS, for atypical diderms, we only identified the common parts between their pathways and the true diderms-LPS pathways. The rest of the true diderms-LPS pathways should have an equivalent in the atypical diderms pathways but our approach by candidate genes did not allow us to identify them. This hints at the possibility of a different evolution from a common base, as some of the functions performed by the genes present in pattern 3 (true diderms-LPS) but absent in pattern 1 (atypical diderms) should be carried out in one way or another (e.g., the maintenance of the outer membrane or the outer membrane invagination during cell division) [110]. In this case, the common base would be the partial Lpt and Tol-Pal systems, upon which at least two different systems for handling the outer membrane would have built in the true diderms-LPS and (all or some) atypical diderms. On the other hand, if the LBCA was a diderm, then extant monoderms would have been the result of several independent secondary simplifications. Consequently, the monoderms dispersed within the Terrabacteria group would share the same origin, a diderm ancestor, but would not necessarily end up with the same remaining genes after their respective simplification. Yet, they all display the same single pattern (pattern 1).

Assuming a monoderm LBCA, single-gene trees might suggest that some outer membrane genes found in atypical diderms (e.g., LptF and LptG) stem from horizontal transfer from true diderms-LPS, rather than through vertical inheritance from a diderm LBCA ancestor. However, because most of these trees are poorly resolved (despite good multiple sequence alignments), the evidence is weak at best. Based on a parsimony reasoning, the exclusivity of pattern 3 to true diderms-LPS and the fact that it is shared between all of them suggest, alongside their well-supported branch in our phylogenomic tree, the monophyly of the true diderms-LPS group. Indeed, if all current genomes of a group have the same subset of genes, the LCA of the group is likely to have had these genes (in a form or another). If correct, the bacterial root cannot lie within true diderms-LPS and as already mentioned, a root on (or within) Terrabacteria implies that the diderm cell-wall architecture appeared at least on two separate occasions. The latter inference is necessary to account for diderms other than true diderms-LPS in Firmicutes, Cyanobacteria, Chloroflexi and Deinococcus-Thermus, which then raises the issue of how the lipopolysaccharides are transported from the IM to the outer membrane for these atypical diderms nested within Terrabacteria. Indeed, they do not share the same Lpt system as true diderms-LPS as theirs is “reduced”, so they must have developed another system grafted (or not) onto the “reduced” Lpt system.

Another clue that might confirm our reconstruction is that the rare organisms amongst the CPR (Candidate Phylum Radiation, also known as Patescibacteria [62,111]) to have been described to feature a monoderm cell-wall architecture [112]. In several trees including the CPR (with the Archaea used as the outgroup), these are the first to diverge from the other bacteria, while the remaining of those trees have the same structure as ours [64,65]. However, in [25,113], the CPR subtree is found within the Terrabacteria with strong support. Consequently, depending on the accepted topology, the CPR could either be another (small) clue for a monoderm LBCA (CPR at the base of the bacterial tree) or only for a monoderm ancestor for the Terrabacteria group (CPR within the Terrabacteria group). Nonetheless, as most CPR genomes still lack detailed reliable information about the cell-wall architecture of the corresponding organisms, there was no point adding them to our study for now.

When it comes to the reconstruction of the *dcw* cluster, the LBCA is inferred as featuring a complete 17-gene cluster. This complete cluster has probably been vertically transmitted since then and often subject to parallel reduction, either by escape of one or several genes from the cluster or by disappearance of those genes from the genome. As it is shared by both monoderm and diderm organisms, the *dcw* cluster does not give a clue about the issue of the number of membranes of the LBCA. However, it confirms that the LBCA had a cell wall with a peptidoglycan layer, even if it does not inform on its original thickness. 

In true diderms-LPS and Terrabacteria, the *murA* gene is (almost) always absent from the main *dcw* cluster. In Firmicutes, which are at the base of Terrabacteria, this gene is nevertheless considered located within the cluster by our reconstruction, as this is the situation for five (out of nine) genomes from our selection of 85 representatives. The gene is also found in sub-clusters distributed relatively patchily across Cyanobacteria, Firmicutes, Epsilon-proteobacteria, Elusimicrobia, *Caldithrix abyssi*, *planctomycete KSU1*, and *Lentisphaera araneosa*. Both extant and reconstructed ancestors show that true diderms-LPS have excised their *murA* from the main cluster after diverging from Terrabacteria, whereas Terrabacteria kept it longer in the main cluster. However, *murA* is found located on sub-clusters in both groups.

For the moment, there is no scenario to explain the appearance of the outer membrane in the lineage leading to true diderms-LPS, but such a scenario exists for the appearance of diderms in Firmicutes: it is the failed endospore origin [11,13,15,114]. According to this hypothesis, an ancestral monoderm endospore former would have experienced a failed sporulation, thereby locking the endospore within the cell while never finishing the spore. With time, it would have become a diderm bacteria. Indeed, during sporulation, the prespore engulfed in the bacterial mother cell has two membranes. A thin layer of the mother peptidoglycan subsists between these membranes before the cortex is added around the prespore between this small layer and the outer membrane. Although not yet a diderm-LPS architecture, a cortex-less spore could be a starting point for the emergence of diderm bacteria in the specific case of Firmicutes. In 2016, Tocheva [14] amended the model by arguing that this founding event would have taken place in an ancestor not only to diderm Firmicutes but to all diderm bacteria. Regarding the origin of the outer membrane in atypical diderms other than Firmicutes, we have already mentioned that Chloroflexi might be monoderms, based on their shared pattern (pattern 2) with monoderm Terrabacteria. This leaves us with Cyanobacteria and Deinococcus-Thermus, along with the large true diderms-LPS group. Because pattern 3 looks like a complexification of pattern 1, the origin of didermia in true diderms-LPS might come from one of these atypical diderms phyla by horizontal gene transfer of outer membrane genes, followed by complexification in an ancestor of true diderms-LPS. Alternatively, true diderms-LPS ancestors might have transferred outer membrane genes to distinct ancestors of atypical diderms phyla, thus in the opposite direction. At this stage, this remains an open question because of the lack of resolution of the corresponding single-gene trees, which prevents any definitive answer. However, it is of note that the failed sporulation scenario is compatible with the inferences of [105].

## 5. Conclusions

Our results suggest that the LBCA might have been, against familiar parsimony reasoning, a monoderm bacteria with a thick peptidoglycan layer, which is also supported by the recent study of [105]. The reconstruction of the *dcw* cluster adds a strong hint towards an LBCA with a peptidoglycan layer but does not discriminate between a thick and a thin peptidoglycan layer. Concerning our study of the outer membrane genes, their distribution suggests that indeed a monoderm ancestor is possible, but the evidence is not decisive. Yet, further improving our results using the same methods would require a more accurate description of the cell-wall architecture of the extant organisms, notably the presence or absence of the lipopolysaccharides, an information which, in our experience, is often lacking. When available, it is concentrated in the older literature, when organisms were cultivated and characterized before being sequenced, in contrast to the numerous candidate bacterial phyla that populate recent phylogenomic trees [66,67]. Nevertheless, even older genomes do not guarantee an exploitable description, like *Rivularia* sp. (Appendix A: 38). Moreover, we observe that some outer membrane genes involved with the precursors of lipopolysaccharides synthesis are also present in genomes of bacteria that does not have lipopolysaccharides on their outer membrane (or even an outer membrane), thus relying solely on the presence of specific genes to determine the presence or absence of lipopolysaccharides is not adequate.

One could argue that the current study does not concern the LBCA but the LCA of cultured (and characterized) Bacteria and we would not completely disagree as we ourselves see it as a proof of concept of the method. A follow-up would be interesting to carry out once accurate information for the cell wall of more phyla are available. In such a follow-up study, it could be interesting to add supplementary genomes such as the “rogue” lineages (e.g., Aquificae and Thermotogae), additional phyla of uncertain phylogenetic position (e.g., basal Terrabacteria), completely new genomes (e.g., CPR) or even an outgroup to root the tree (e.g., Archaea). Aquifex being “just another’’ group of diderms and Thermotogae being a chimera with a specific diderm architecture, their inclusion would only provide a limited amount of information compared to considering additional Terrabacteria genomes or representatives of the recently discovered CPR. Regarding the difficulty to place such lineages accurately in a phylogenomic tree, it could be overcome by adding genes that are not single copy but at the expense of more work to sort out the orthologous copies. The CPR group would be a particularly welcome addition, provided a useful description of their cell wall could be obtained. Concerning the addition of an outgroup, the question of how it will be used should be answered first: will it be included in the cell-wall reconstruction analyses or will it only be used to root the bacterial subtree. Indeed, if it is not used for reconstruction, any slow evolving fully sequenced Archaea would be usable. On the other hand, if we are interested in reconstructing their cell wall too, we would need to select them very carefully, just as we did for Bacteria. In this respect, the cell-wall diversity of Archaea is as complicated as the bacterial one, if not more, which would add another level of difficulty, and thus uncertainty, to the inferred results.

## Figures and Tables

**Figure 1 genes-13-00376-f001:**
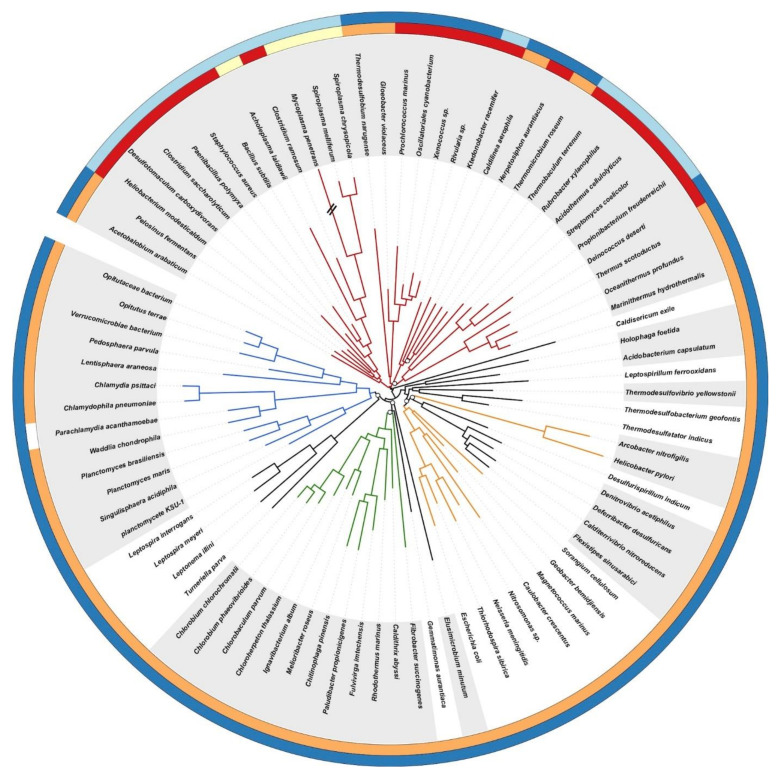
Phylogenomic tree of the bacterial domain based on a supermatrix concatenating 117 single-copy orthologous genes chosen for their broad conservation across Bacteria. The tree was rooted on Terrabacteria. The supermatrix had 85 species and 19,959 unambiguously aligned amino-acid positions (<5% missing character states). The tree was inferred from amino-acid sequences using PhyloBayes MPI and the CAT+GTR+Γ model of sequence evolution. Open symbols at the nodes are posterior probabilities (PP), and nodes without a symbol correspond to maximum statistical support for phylogenetic inference (posterior probabilities of 1.0; averaged over two MCMC chains). The length of the branch marked with “//” has been reduced by 50% for the sake of clarity. Colour key is red = Terrabacteria, orange = Proteobacteria, green = FBC superphylum, blue = PVC superphylum. Outer circles stand for the status of the peptidoglycan (PG) and of the outer membrane in the organisms, according to our literature survey: red = thick PG, orange = thin PG, yellow = no PG, dark blue = diderm, light blue = monoderm, white = no information. Alternating white and grey backgrounds highlight the alternance between differentially coloured groups or phyla.

**Figure 2 genes-13-00376-f002:**
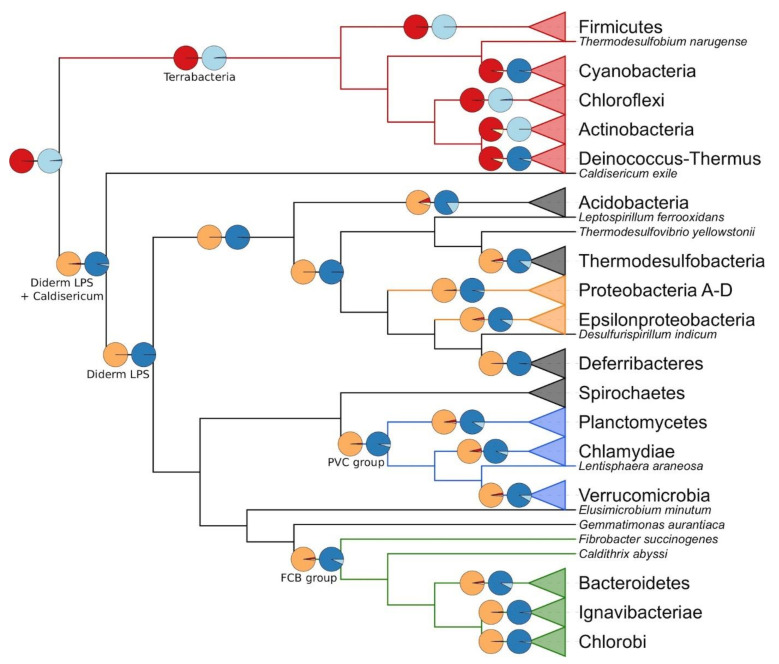
Cladogram derived from the tree of Figure 1 featuring the cell-wall architecture inferred for selected last common ancestors among Bacteria. Colour key is red = Terrabacteria, orange = Proteobacteria, green = FBC superphylum, blue = PVC superphylum Branches ending with a triangle represent collapsed groups (for details, see Figure 1 or Appendix A). The pie chart sectors correspond to the posterior probabilities of the model reverse-jump hyperprior exponential 0 to 100 (R2). Colour key is red = thick PG, orange = thin PG, yellow = no PG, dark blue = diderm, light blue = monoderm.

**Figure 3 genes-13-00376-f003:**
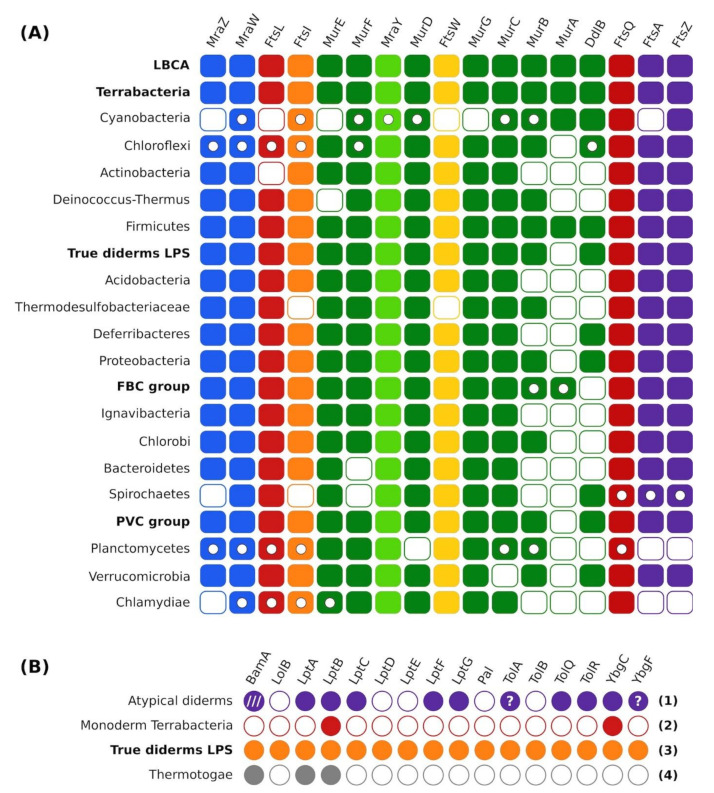
Overview of gene distribution and synteny analyses. (**A**) ProCARs results for *dcw* cluster organization in selected LCA among Bacteria. Full rectangle = gene present and in the main cluster; empty circle in rectangle = gene present but in a sub-cluster; empty rectangle = gene present but outside of any cluster. Note that the reconstruction procedure prevents the complete lack of a gene in an ancestral genome. (**B**) Recurring distribution patterns at the phylum level for the proteins involved with the outer membrane. Full circle = gene present in the group; empty circle = gene absent in the group; “?” in a circle = potential presence of the gene in the group; /// = presence in a sub-group only (i.e., Deinococcus-Thermus). Numbers in bold are the pattern numbers. Names written in bold are the names of groups regrouping several phyla.

**Table 1 genes-13-00376-t001:** Overview of BayesTraits results. qij design posterior transition rates, whereas P(i) correspond to posterior ancestral state probabilities. For the membrane (MBN) trait, state 0 = one MBN and state 1 = two MBN, while for the peptidoglycan (PG) trait, state 0 = no PG, state 1 = thin PG and state 2 = thick PG. “H biased” is the model where the hyperprior has been purposely biased to favour a diderm LBCA (see Materials and Methods for details).

Node	Trait	Statistic	E	H1	H2	R1	R2	H Biased
LBCA	MBN	mean q01	2.495	2.352	2.477	2.288	2.411	1.431
LBCA	MBN	mean q10	0.132	0.113	0.121	0.012	0.008	0.210
LBCA	MBN	mean P(0)	94.951	94.204	95.375	97.134	98.161	67.092
LBCA	PG	mean P(0)	22.068	4.022	38.604	0.397	0.594	N/A
LBCA	PG	mean P(2)	76.497	94.622	60.147	99.535	99.358	N/A
LBCA	PG	mean q01	4.626	1.634	7.317	0.798	0.827	N/A
LBCA	PG	mean q02	6.935	2.020	20.967	0.953	1.041	N/A
LBCA	PG	mean q10	0.166	0.102	0.187	0.000	0.000	N/A
LBCA	PG	mean q12	0.128	0.109	0.118	0.001	0.000	N/A
LBCA	PG	mean q20	2.088	0.937	4.941	1.347	1.413	N/A
LBCA	PG	mean q21	1.890	2.165	1.600	1.398	1.419	N/A
Firmicutes	PG	mean P(0)	17.631	3.936	30.120	0.611	0.738	N/A
Firmicutes	PG	mean P(2)	81.891	95.648	69.435	99.378	99.237	N/A

**Table 2 genes-13-00376-t002:** Pairwise comparisons of BayesTraits model fit using Bayes Factors (BF). BF > 2 are interpreted as positive evidence, 5 ≤ BF < 10 as strong evidence and BF > 20 as very strong evidence in favour of the more complex model [89].

Complex	Simple	MBN	PG
R1	H2	7.41	22.86
	E	5.95	17.47
	H1	2.69	8.38
	R2	2.42	1.91
R2	H2	4.99	20.95
	E	3.53	15.56
	H1	0.27	6.47
H1	H2	4.71	14.47
	E	3.25	9.09
E	H2	1.46	5.39

## Data Availability

The datasets generated and analysed for this study can be found in the FigShare repository available here: https://doi.org/10.6084/m9.figshare.14932386.v2 (16 February 2022). Similarly, the database schema and corresponding table dump are available at: https://doi.org/10.6084/m9.figshare.17102651.v1 (16 February 2022).

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
