# Peer review of "Was the Last Bacterial Common Ancestor a Monoderm after All?"

_genes, 2022, doi:10.3390/genes13020376_

Round 1
Reviewer 1 Report
In this article entitled “Was the last bacterial common ancestor a monoderm after all?”, Léonard et al. present an analysis of bacterial cell envelope evolution and propose that the last bacterial common ancestor (LBCA) was a monoderm. First, they predicted the nature of the cell envelope in the deep branches of the tree of Bacteria but also at the ancestor using Bayesian approaches. Then, they describe the evolution of DCW and LPS cluster genes. Finally, they discuss about their results and conclude that the LBCA was likely monoderm. The study of the nature of the cell envelope in the ancestor of Bacteria is of particular interest. However, strong methodological issues impair heavily the quality of the results and make the conclusions not reliable. Also, the DCW cluster section does not appear essential to the main question, the discussion is long and hard to follow and some statements are not supported either by the data or the literature.
1. My major concern is about the phylogenetic tree of the 85 Bacteria that the author used to infer the ancestral states (Fig. 1 and 2). The tree does not cover the currently known diversity of phyla, especially for Terrabacteria. Indeed, the authors only selected five major phyla within Terrabacteria, but there are many more that have been described (Melainabacteria, Dictyoglomi, Armatimonadetes, Aerophobetes, Wallbacteria, Synergistetes, …). Importantly, the majority of the missing lineages are (or have been predicted to be) diderm. Therefore, their absence and particularly the lack of basal lineages of Terrabacteria likely strongly impacts the inference of the ancestral states. This taxonomic undersampling also likely introduces a strong bias in phylogenetic inference leading to the apparent misplacement of major phyla. Indeed, the tree that is presented in Fig. 1 and 2 presents major discrepancies with recent published phylogenies (Coleman et al., 2021, Castelle et al., 2018), such as the placement of Firmicutes at the base of the Terrabacteria which has never been observed. Also, within the group of Firmicutes, the diderm basal lineages are poorly represented (only one Halanaerobiales, no Limnochordia). This probably strongly affected the inference of a monoderm ancestor in the Firmicutes.
As the group of Gracillicutes includes only diderms and Terrabacteria a mix of diderms and monoderms, the ancestral state of LBCA but also the rates of transitions depend mainly on the group of Terrabacteria. In this context, a more rigorous analysis of this group by including more phyla is crucial to reach reliable conclusions. I understand that the authors wanted to keep only lineages for which experimental data on the cell envelope is available but this stringent selection precludes a rigorous phylogenetic analysis. The presence of outer membrane markers such as LPS genes can be rather safely used to infer a diderm state, even if indirect evidence.
2. Another concern is about the DCW cluster description section. It is indeed interesting to study the evolution of the DCW cluster to highlight some patterns that could distinguish monoderms and diderms. Nevertheless, the results presented do not bring any element on the main question of whether the LBCA was a diderm or a monoderm. In fact, the authors did not find any specific pattern, so this part is to me dispensable for the paper (or at least should be strongly reduced). Instead, the study of the LPS genes seems to be more interesting as a clear pattern between monoderm and diderm is spotted. Nevertheless, the analysis is not deep enough to provide key data to answer to the main question. The authors just describe the distribution and very rapidly the phylogenies, with no further analysis.
3. The use of “true diderm” for Gracilicutes is very misleading. It suggests that all diderms from Terrabacteria are not “true” diderms, which is false. Indeed, “true” diderms are also present in Terrabacteria, although some are atypical (Coleman et al., 2021). It seems that the authors minimize the nature of diderms in Terrabacteria to fit with their hypothesis.
4. The fact that Thermotoga have a high rate of genes originating from HGTs in their genomes is to me not a sufficient reason to remove them from the analysis. The author removed another key diderm phylum from Terrabacteria, again likely biasing the results in favor of a monoderm LBCA.
5. Some other statements from the authors are not supported by the data or literature:
-lines 663-665: what are the evidences for an HGT of BamA from Gracilicutes to Deinococcus-Thermus and Thermotogae?
-lines 670-671: An atypical architecture of BamA in Cyanobacteria and Firmicutes is not a sufficient reason to remove it from the dataset. It happens quiet often that some orthologues vertically inherited fuse with other domains. Here, only a solid phylogenetic analysis of the entire family can highlight a potential complex history with HGTs justifying its removal.
-lines 716-717: Fig. 3B clearly shows that LPS gene distribution correlates with the presence of an outer membrane. Furthermore, we know that LPS synthesis and transport are tightly linked to the outer membrane. In this context, the reasoning that the absence of some components of LPS transport in some diderms indicates that LPS transport could have a function independent from the outer membrane seems not very likely. A more logical explanation could be that the missing components have been replaced by others. It seems again an attempt to minimize the diderm nature of Terrabacteria.
-lines 724-726: The claim that lpt genes are only clustered in the genomes of Proteobacteria is in contradiction with recent literature (Taib et al., 2020). For example, some Terrabacteria such as Cyanobacteria, Firmicutes or Armatimonadetes exhibit clusters with various combinations of lptABCF in their genomes.
-lines 789-792: The presence of two LPS genes, lptB and ybgC, in monoderm lineages may indeed indicate a broader role than LPS transport. Nevertheless, the other 9-14 genes seem specific to diderms, so the peculiar case of LptB and YbgC is not a sufficient reason to preclude an analysis of the entire set of LPS genes. Furthermore, LptB belong to a very large ATPase family. Thus, the detection of homologues in all phyla may be due to a weak specificity during the homologue research.
-line 811: The authors consider some Chloroflexi as diderms, and state that even if they are monoderm, the inference of LBCA state is not expected to change. From this, they hypothesize that this is another evidence for the monoderm LBCA nature. In fact, the state of Chloroflexi has just no influence on the LBCA state, so it does not add any evidence on the diderm or monoderm nature of the LBCA.
-lines 840-852: This part is very unclear. Also, how a difference in term of pattern between Gracilicutes and Terrabacteria diderms can indicate that Gracilicutes are monophyletic is not understandable.
Author Response
The reviewer's comments are in italic style, our answers in plain style and introduced by three carets: >>>
Moderate English changes required
>>> We have made a number of minor changes to improve the reading of text, deleting unnecessary adverbs and using commoner synonyms when possible.
In this article entitled “Was the last bacterial common ancestor a monoderm after all?”, Léonard et al. present an analysis of bacterial cell envelope evolution and propose that the last bacterial common ancestor (LBCA) was a monoderm. First, they predicted the nature of the cell envelope in the deep branches of the tree of Bacteria but also at the ancestor using Bayesian approaches. Then, they describe the evolution of DCW and LPS cluster genes. Finally, they discuss about their results and conclude that the LBCA was likely monoderm. The study of the nature of the cell envelope in the ancestor of Bacteria is of particular interest. However, strong methodological issues impair heavily the quality of the results and make the conclusions not reliable. Also, the DCW cluster section does not appear essential to the main question, the discussion is long and hard to follow and some statements are not supported either by the data or the literature.
>>> Thank you for this mostly accurate assessment of our work. We would like to insist on the fact that we do not advocate a monoderm state for the LBCA but rather cautiously report that objective analyses, even though not perfect, point towards this possibility. We do not over-interpret our results and explicitly acknowledge the potential shortcomings of our data Here are some examples of our cautious wording:
- From our abstract: “Quite unexpectedly, our analyses suggest that all extant bacteria might have evolved from a common ancestor with a monoderm cell-wall architecture. If true, this indicates that the appearance of the outer membrane was not a unique event and that selective forces have led to the repeated adoption of such an architecture.”
- From our results: “Regarding the presence of the bamA gene in the atypical diderms of the group Deinococcus-Thermus, it has already been reported [96] and this result appears less compatible with a monoderm LBCA.”
- From our conclusion: “​​Our results suggest that the LBCA might have been, against common parsimony reasoning, a monoderm bacteria with a thick peptidoglycan layer. The reconstruction of the dcw cluster adds a strong hint towards an LBCA with a peptidoglycan layer but does not discriminate between a thick and a thin peptidoglycan layer. Concerning our study of the outer membrane genes, their distribution suggests that indeed a monoderm ancestor is possible, but the evidence is not decisive.”
Granted, our conclusions are at odds with previous (some very recent) studies, but these studies were based on other (often, yet not always, inferior) approaches. Thus, our work is certainly worth publishing as “food for thought”, even if future research starting from more primary data may eventually prove us wrong. In our opinion, this is the essence of research in evolutionary biology: many studies address the same question and the gradual improvement in both data collection and analytical methods progressively shape our view. Regarding the reliability and limitations of our approaches, we see our study as a proof of concept. We will need more information (and also more accurate information) about extant cell-wall architectures than what is available right now to go further down this path. This point is addressed in the conclusion (pages 21-22).
My major concern is about the phylogenetic tree of the 85 Bacteria that the author used to infer the ancestral states (Fig. 1 and 2). The tree does not cover the currently known diversity of phyla, especially for Terrabacteria. Indeed, the authors only selected five major phyla within Terrabacteria, but there are many more that have been described (Melainabacteria, Dictyoglomi, Armatimonadetes, Aerophobetes, Wallbacteria, Synergistetes, …). Importantly, the majority of the missing lineages are (or have been predicted to be) diderm. Therefore, their absence and particularly the lack of basal lineages of Terrabacteria likely strongly impacts the inference of the ancestral states. This taxonomic undersampling also likely introduces a strong bias in phylogenetic inference leading to the apparent misplacement of major phyla. Indeed, the tree that is presented in Fig. 1 and 2 presents major discrepancies with recent published phylogenies (Coleman et al., 2021, Castelle et al., 2018), such as the placement of Firmicutes at the base of the Terrabacteria which has never been observed. Also, within the group of Firmicutes, the diderm basal lineages are poorly represented (only one Halanaerobiales, no Limnochordia). This probably strongly affected the inference of a monoderm ancestor in the Firmicutes.
>>> It depends on what you want to achieve. If your goal is to get a picture of the current bacterial diversity based on genomes and metagenome-assembled genomes, it is indeed useful to sample all known phyla, including those for which no (or only very partial) phenotypic data is available (but be very careful to genome contamination, see Lupo et al. 2021). Several recent studies have tried to do precisely that (e.g., Parks et al., 2018; Zhu et al., 2019; Cavalier-Smith et al., 2020). In our case, we wanted to trace the evolution of the cell-wall architecture across bacteria. Obviously, we needed reliable phenotypic data (not predicted cell-wall architectures), which mechanically reduced the amount of phyla that were possible to consider (please have a look at the reference list in Table S4 to get an idea of the effort required just for the 85 organisms that we studied in detail). Again, this potential limitation of our study is addressed in the conclusion (pages 21-22), but we now underline in the text that we have not studied all possible phyla.
- Parks, Donovan H., et al. "A standardized bacterial taxonomy based on genome phylogeny substantially revises the tree of life." Nature Biotechnology 36.10 (2018): 996-1004.
- Zhu, Qiyun, et al. "Phylogenomics of 10,575 genomes reveals evolutionary proximity between domains Bacteria and Archaea." Nature Communications 10.1 (2019): 1-14.
- Cavalier-Smith, Thomas, E. Ema, and Yung Chao. "Multidomain ribosomal protein trees and the planctobacterial origin of neomura (eukaryotes, archaebacteria)." Protoplasma 257.3 (2020): 621-753.
- Lupo, Valérian, et al. "Contamination in Reference Sequence Databases: Time for Divide-and-Rule Tactics." Frontiers in Microbiology 12 (2021).
As the group of Gracillicutes includes only diderms and Terrabacteria a mix of diderms and monoderms, the ancestral state of LBCA but also the rates of transitions depend mainly on the group of Terrabacteria. In this context, a more rigorous analysis of this group by including more phyla is crucial to reach reliable conclusions. I understand that the authors wanted to keep only lineages for which experimental data on the cell envelope is available but this stringent selection precludes a rigorous phylogenetic analysis. The presence of outer membrane markers such as LPS genes can be rather safely used to infer a diderm state, even if indirect evidence.
>>> We see your point. It makes sense but we have a different perception. We wanted to use an array of different evidence applicable (at least) to the same selection of 85 organisms, rather than betting on indirect evidence available for more genomes. Yet, we do it slightly for the outer membrane (OM) genes, as they were mined in 903 genomes, including additional phyla not considered in the synteny (dcw gene cluster) and cell-wall traits analyses.
By the way, we contacted Taib and collaborators several times to get the phenotypic data behind their large study of cell-wall evolution in Firmicutes and apply our methodology to it, but they were unable to provide it to us. We hypothesize that such data is not actually available in a form that would be exploitable in a Bayesian framework (i.e. reliable character states on an individual genome basis).
- Taib, Najwa, et al. "Genome-wide analysis of the Firmicutes illuminates the diderm/monoderm transition." Nature Ecology & Evolution 4.12 (2020): 1661-1672.
Another concern is about the DCW cluster description section. It is indeed interesting to study the evolution of the DCW cluster to highlight some patterns that could distinguish monoderms and diderms. Nevertheless, the results presented do not bring any element on the main question of whether the LBCA was a diderm or a monoderm. In fact, the authors did not find any specific pattern, so this part is to me dispensable for the paper (or at least should be strongly reduced). Instead, the study of the LPS genes seems to be more interesting as a clear pattern between monoderm and diderm is spotted. Nevertheless, the analysis is not deep enough to provide key data to answer to the main question. The authors just describe the distribution and very rapidly the phylogenies, with no further analysis.
>>> The dcw gene cluster might have been relevant to the question of the diderm/monoderm ancestry. For example, it could have helped us to root the bacterial tree (i.e., polarize evolution), considering that purely molecular evidence is susceptible to lead to phylogenetic artifacts in such a case (Gouy et al., 2015). That is why we investigated dcw gene cluster evolution (among many other things, as you correctly recall). However, we agree with you that the outcome is not that exciting, at least not beyond confirming that peptidoglycan is ancestral to all extant bacteria and the early departure of the murA gene from the dcw gene cluster. Indeed, in almost every phyla studied, murA is absent from the cluster, but not the genome. This means at least two losses, one for the Gracilicutes LCA and one for the Chloroflexi-Actinobacteria-Deinococcus-Thermus LCA, or up to four losses if the Chloroflexi, Actinobacteria and Deinococcus-Thermus dcw clusters have lost murA independently. We have added a sentence about murA to better make use of the data.
- Gouy, Richard, Denis Baurain, and Hervé Philippe. "Rooting the tree of life: the phylogenetic jury is still out." Philosophical Transactions of the Royal Society B: Biological Sciences 370.1678 (2015): 20140329.
That being said, Genes’ format is not the one of a Nature or Science paper: it allows reporting on “dull” results, provided that the methodology is sound (and in our case the question is moreover interesting). We thus feel that dcw analyses are not out of place in our study and are inclined to keep them.
The use of “true diderm” for Gracilicutes is very misleading. It suggests that all diderms from Terrabacteria are not “true” diderms, which is false. Indeed, “true” diderms are also present in Terrabacteria, although some are atypical (Coleman et al., 2021). It seems that the authors minimize the nature of diderms in Terrabacteria to fit with their hypothesis.
>>> We see your point but this was not our intention. We never use the expression “true diderms” without the “LPS” addition. When we mention TDL (true diderms-LPS), we always refer to the monophyletic group corresponding to Gracilicutes. Since the latter taxon is not that widespread in the literature (e.g. it is still missing from NCBI Taxonomy), TDL is likely to be better understood by the reader: a bacterium with a thin peptidoglycan layer and a standard LPS-containing outer membrane (i.e. a “Gram-negative”), except some clearly derived organisms such as the Chlamydia. For example, in Terrabacteria, Cyanobacteria are indeed diderm, but their peptidoglycan layer is thicker.
The fact that Thermotoga have a high rate of genes originating from HGTs in their genomes is to me not a sufficient reason to remove them from the analysis. The author removed another key diderm phylum from Terrabacteria, again likely biasing the results in favor of a monoderm LBCA.
>>> Yes, and we argue this was a sound decision. If one cannot position some lineages in the reference tree, even when using very powerful evolutionary models (here, CAT+GTR), it is better to discard them to avoid biasing the biological conclusions. This is precisely in such an area that one can see that we were cautious and thorough when carrying out this research, rather than trying to “show-off” unreliable results based on a plethora of misplaced organisms. Furthermore, the cell-wall architecture of Thermotoga is specific to them; thus, it would have been encoded differently than the regular diderm state and would likely have had little influence on the outcome of the BayesTraits reconstruction.
Some other statements from the authors are not supported by the data or literature:
-lines 663-665: what are the evidences for an HGT of BamA from Gracilicutes to Deinococcus-Thermus and Thermotogae?
>>> This hypothesis is not stated in our study. We simply mentioned the existence of the classical/true BamA in the Gracilicutes, Thermotoga and Deinococcus-Thermus, as reported in Heinz & Lithgow (2014) but we did not discuss the origin of BamA in Thermotogae and Deinococcus-Thermus. However, we state that, due to the complexity of the Omp85/TpsB family (to which BamA belongs), it would certainly be an interesting subject for a new study (see our answer to your next comment just below).
- Heinz, Eva, and Trevor Lithgow. "A comprehensive analysis of the Omp85/TpsB protein superfamily structural diversity, taxonomic occurrence, and evolution." Frontiers in Microbiology 5 (2014): 370.
-lines 670-671: An atypical architecture of BamA in Cyanobacteria and Firmicutes is not a sufficient reason to remove it from the dataset. It happens quiet often that some orthologues vertically inherited fuse with other domains. Here, only a solid phylogenetic analysis of the entire family can highlight a potential complex history with HGTs justifying its removal.
>>> In our opinion, it is exactly the opposite: one needs a solid domain-based (not full-length) phylogenetic analysis to demonstrate that all (or even most) BamA-like (sensu lato) proteins can be considered as orthologues. Any alteration to the domain architecture in such an important gene (for the outer membrane) is a potential hint for more complex history, including domain shuffling and horizontal gene transfer, even if only once or twice. Here, we did not exclude anything; we simply focused on the orthologous groups including the typical BamA sequences to build our HMM profile. For an example of what we mean by a domain-based phylogenetic analysis, see Califice et al. (2012). And yes, we would be happy to have a closer look at a very carefully built BamA tree (especially from the point of view of sequence sampling and orthology assessment). Actually, we have started to build such a tree, but the complexity of the task is beyond the scope of the current manuscript.
- Califice, Sophie, et al. "A single ancient origin for prototypical serine/arginine-rich splicing factors." Plant Physiology 158.2 (2012): 546-560.
-lines 716-717: Fig. 3B clearly shows that LPS gene distribution correlates with the presence of an outer membrane. Furthermore, we know that LPS synthesis and transport are tightly linked to the outer membrane. In this context, the reasoning that the absence of some components of LPS transport in some diderms indicates that LPS transport could have a function independent from the outer membrane seems not very likely. A more logical explanation could be that the missing components have been replaced by others. It seems again an attempt to minimize the diderm nature of Terrabacteria.
>>> We only hypothesize alternative functions for two of the 16 OM-related genes studied in our manuscript, lptB and ybgC, because these are present in monoderms too, not only diderms, whether TDL or not. Regarding ybgF, there is already a proposal for an alternative function. Of course, we discuss these results in the context of our BayesTraits reconstruction, but we nevertheless acknowledge that they are not conclusive (section 3.4, page 17) and we also use the same explanation as you for the partial presence of the OM-related genes in atypical diderms (replaced components, Discussion, page 20).
-lines 724-726: The claim that lpt genes are only clustered in the genomes of Proteobacteria is in contradiction with recent literature (Taib et al., 2020). For example, some Terrabacteria such as Cyanobacteria, Firmicutes or Armatimonadetes exhibit clusters with various combinations of lptABCF in their genomes.
>>> You are correct. We were not precise enough concerning the basis of our claim about lpt and tol-pal genes. We have changed the text to add a precision (page 18): the study of the synteny of the lpt genes was conducted only in the 85 genomes (including 10 genomes of Firmicutes) of the phylogenomic tree and not in the 903 genomes. This could explain that we missed some information while Taib et al. 2020, by studying one Proteobacteria and twelve Firmicutes among the 1639 Firmicutes genomes they consider in their work (of which five belong to two Firmicutes groups with an outer membrane), did not.
- Taib, Najwa, et al. "Genome-wide analysis of the Firmicutes illuminates the diderm/monoderm transition." Nature Ecology & Evolution 4.12 (2020): 1661-1672.
-lines 789-792: The presence of two LPS genes, lptB and ybgC, in monoderm lineages may indeed indicate a broader role than LPS transport. Nevertheless, the other 9-14 genes seem specific to diderms, so the peculiar case of LptB and YbgC is not a sufficient reason to preclude an analysis of the entire set of LPS genes. Furthermore, LptB belong to a very large ATPase family. Thus, the detection of homologues in all phyla may be due to a weak specificity during the homologue research.
>>> We did not preclude any analysis. We studied the distribution of the 16 OM-related genes based on e-value/hit length/profile-coverage filtered HMM searches using HMM profiles built from manually curated alignments corresponding to one to n orthologous groups identified as having one of the 16 OM-related genes. This is already a quite important endeavor for a subsection of a manuscript.
In more detail, here is what we did about LptB and the other OM-related proteins. We first identified sequences annotated as the proteins of interest (or clustered in orthologous groups with sequences annotated as the proteins of interest) after OrthoMCL (Li et al., 2003). They were aligned and the alignment then transformed into an HMMER profile (Potter et al., 2018), often combining multiple files based on preliminary trees. Using this HMMER profile onto a database representing the complete proteomes of our 903 genomes, we filtered the HMM hits with the GUI of ompa-pa (Bertrand and Baurain, unpublished): up to +/- 100 AAs compared to the E.coli length (less for the smaller proteins) and with at least a profile coverage of 0.7 (this ensures architecture conservation and not only domain matching). We may have been too prudent but we think that stringent thresholds are more likely to yield proteins that are clearly related to the protein of interest and not distant homologues that only share some similarity (as you advocate yourself in your comment). In the case of LptB, we might have captured some more distant sequences but we argue that our selection had at worst a few of them since, LptB being a small protein (241 AAs), we used a smaller size range (+/- 50 AAs; between 200-300 AAs) and still enforced profile coverage.
- Li, Li, Christian J. Stoeckert, and David S. Roos. "OrthoMCL: identification of ortholog groups for eukaryotic genomes." Genome research 13.9 (2003): 2178-2189.
- Potter, Simon C., et al. "HMMER web server: 2018 update." Nucleic acids research 46.W1 (2018): W200-W204.
- ​​https://metacpan.org/dist/Bio-MUST-Apps-OmpaPa
-line 811: The authors consider some Chloroflexi as diderms, and state that even if they are monoderm, the inference of LBCA state is not expected to change. From this, they hypothesize that this is another evidence for the monoderm LBCA nature. In fact, the state of Chloroflexi has just no influence on the LBCA state, so it does not add any evidence on the diderm or monoderm nature of the LBCA.
>>> In the manuscript, we interpret this non-effect of “diderm” Chloroflexi as a sign of a strong signal towards a monoderm ancestor, a signal that we did not manage to overcome, even when tweaking the priors to strongly bias them in favor of a diderm cell wall. Regarding our analyses of the dcw gene cluster and the OM-related genes Chloroflexi did not confirm nor infirm the monoderm nature of the reconstructed LBCA. Finally, we also argue that our encoding of the Chloroflexi is a small evidence and thus not decisive by itself.
-lines 840-852: This part is very unclear. Also, how a difference in term of pattern between Gracilicutes and Terrabacteria diderms can indicate that Gracilicutes are monophyletic is not understandable.
>>> We apologize for the lack of clarity here. We have modified the text to better explain our reasoning (page 20). In the phylogenomic tree, the TDL/Gracilicutes group is statistically well supported. Moreover, all members of this group share the same gene pattern for the OM-related genes (they have almost all genes) but, outside of this group, no other genome presents the same pattern. A parsimonious reasoning to explain how all these genomes share the same pattern would be that the LCA of the TDL group also presented this pattern and vertically transmitted these genes to its descendants. Thus, this observation independently suggests the monophyly of the TDL/Gracilicutes group.
Reviewer 2 Report
Minor Comment
1) Abstract – Line 21 Comma was missing in sentence “Finally extensive similarity searches were carried out to determine the phylogenetic distribution of the genes involved with the biosynthesis of the outer membrane in diderm bacteria.”
Overall:
The author has very discussed about the background information last bacterial common ancestor particularly the cell wall and clearly indication the experimental plan to prove his hypothesis to be true. Further, mentions about the usage of Bayesian interference, homology- and model-based methods to create a phylogenomic tree to characterize the cell wall architecture.
Articles supporting this hypothesis have been cited in this article such as “Genome-wide analysis of the Firmicutes illuminates the diderm/monoderm transition”, which talks about transition between a monoderm and diderm during evolution of bacteria and “The metabolic network of the last bacterial common ancestor” which predicts core for metabolic network of the LBCA and many more articles.
Author Response
English language and style are fine/minor spell check required
Minor Comment
1) Abstract – Line 21 Comma was missing in sentence “Finally extensive similarity searches were carried out to determine the phylogenetic distribution of the genes involved with the biosynthesis of the outer membrane in diderm bacteria.”
>>> Done. Thank you.
Overall:
The author has very discussed about the background information last bacterial common ancestor particularly the cell wall and clearly indication the experimental plan to prove his hypothesis to be true. Further, mentions about the usage of Bayesian interference, homology- and model-based methods to create a phylogenomic tree to characterize the cell wall architecture.
Articles supporting this hypothesis have been cited in this article such as “Genome-wide analysis of the Firmicutes illuminates the diderm/monoderm transition”, which talks about transition between a monoderm and diderm during evolution of bacteria and “The metabolic network of the last bacterial common ancestor” which predicts core for metabolic network of the LBCA and many more articles.
>>> Thank you. These works are indeed cited in our manuscript.
Round 2
Reviewer 1 Report
I thank the authors for their answer. I acknowledge the efforts that were put in their analysis, and I agree that scientific results should be communicated to the readership even if not conclusive. However, this was not my major concern.
As I already explained, my major concern is that the methodological approach used by Leonard et al. makes the results not trustable. The authors claim that it is justified to discard genomes for which no experimental data is available because it could introduce a bias. However, by doing so the authors are introducing an even stronger and worse bias, as lacking most currently available phyla in Terrabacteria has likely enormous consequences both on the topology of the phylogenetic tree and the inference of ancestral states. As the main result of the paper is the inference of the LBCA cell envelope nature, I maintain that, in its present state, this analysis is not robust enough to be communicated to the scientific community, and the results would be highly misleading.
Finally, I do not understand how this analysis constitutes a proof of concept, also considering that these kind of analyses (reconstruction of ancestral traits) are widely used in the scientific community.
Unless the authors are willing to show that their ancestral state inference is unchanged when using a proper taxonomic sampling.
Author Response
I thank the authors for their answer. I acknowledge the efforts that were put in their analysis, and I agree that scientific results should be communicated to the readership even if not conclusive. However, this was not my major concern.
As I already explained, my major concern is that the methodological approach used by Leonard et al. makes the results not trustable. The authors claim that it is justified to discard genomes for which no experimental data is available because it could introduce a bias. However, by doing so the authors are introducing an even stronger and worse bias, as lacking most currently available phyla in Terrabacteria has likely enormous consequences both on the topology of the phylogenetic tree and the inference of ancestral states. As the main result of the paper is the inference of the LBCA cell envelope nature, I maintain that, in its present state, this analysis is not robust enough to be communicated to the scientific community, and the results would be highly misleading.
Finally, I do not understand how this analysis constitutes a proof of concept, also considering that these kind of analyses (reconstruction of ancestral traits) are widely used in the scientific community.
Unless the authors are willing to show that their ancestral state inference is unchanged when using a proper taxonomic sampling.
>>> We somehow agree with Reviewer 1. That is why we explain in our conclusion what are the limits of our method: due to the lack of reliable phenotypic data, our results only concern the cultivated organisms with experimentally characterized phenotypes. The corresponding genomes are the oldest genomes available and we indeed lack many newly described genomes. However, a recent study by Xavier et al. (2021, Communications Biology) has also chosen to restrict the selection to a (quite biased) subset of genomes (anaerobic organisms) in order to predict the metabolic network of the LBCA. Interestingly, this work also supports a monoderm LBCA, even though it was published after the articles of Taib et al. et Coleman et al. Thus, as long as it is justified (and not hidden under the carpet), it seems acceptable to use a limited selection of organisms in such evolutionary studies.
We consider our work as a proof of concept because it shows, even if the dataset is small, that the method is applicable and on that specific problem and can yield good results (BayesTraits probabilities are very clear-cut and robust to alternative model specifications). Unfortunately, we cannot apply our approach with the numerous additional genomes required by Reviewer 1 since reliable descriptions of the cell-wall architecture are lacking for most of the taxonomic groups currently absent from our dataset. Our conclusion is thus quite conservative, limited to what we consider to have enough proof of, i.e., the presence of the peptidoglycan layer in the LCA of cultivated and characterized organisms.
In summary, as we precise the aforementioned limitations in our conclusion, we first wondered how our article could be misleading to the reader. Then, realizing that maybe not all readers will read it up to the conclusion, we modified the abstract to mirror the information added in the previous revision. This should avoid misleading even the most sloppy readers.
- Xavier, Joana C., et al. "The metabolic network of the last bacterial common ancestor." Communications biology 4.1 (2021): 1-10.
- Taib, Najwa, et al. "Genome-wide analysis of the Firmicutes illuminates the diderm/monoderm transition." Nature ecology & evolution 4.12 (2020): 1661-1672.
- Coleman, Gareth A., et al. "A rooted phylogeny resolves early bacterial evolution." Science 372.6542 (2021): eabe0511.